# Multi-task Learning for Heterogeneous multi-source Block-Wise Missing Data

## Abstract

Multi-task learning (MTL) has emerged as an imperative machine learning tool to solve multiple learning tasks simultaneously and has been successfully applied to healthcare, marketing, and biomedical fields. However, in order to borrow information across different tasks effectively, it is essential to utilize both homogeneous and heterogeneous information. Among the extensive literature on MTL, various forms of heterogeneity are presented in MTL problems, such as block-wise, distribution, and posterior heterogeneity. Existing methods, however, struggle to tackle these forms of heterogeneity simultaneously in a unified framework. In this paper, we propose a two-step learning strategy for MTL which addresses the aforementioned heterogeneity. First, we impute the missing blocks using shared representations extracted from homogeneous source across different tasks. Next, we disentangle the mappings between input features and responses into a shared component and a task-specific component, respectively, thereby enabling information borrowing through the shared component. Our numerical experiments and real-data analysis from the ADNI database demonstrate the superior MTL performance of the proposed method compared to a single task learning and other competing methods.

## 1 Introduction

**Motivation.** Many datasets for specific scientific tasks lack sufficient samples to train an accurate machine learning model. In recent decades, multi-task learning (MTL) has become a powerful tool to borrow information across related tasks for improved learning capacity. In addition, data collected for each task might come from multiple sources; for example, clinic notes, medical images, and lab tests are collected for medical diagnosis. The multi-source data brings richer information for each task, potentially enhancing the MTL. However, this also imposes several key challenges. First of all, it is common that observed data sources for each task are heterogeneous, so some blocks (certain data sources for certain tasks) could be entirely missing, termed as a block-wise missing structure in the literature. Second, even if the observed data sources are aligned across tasks, the distribution of the same data source could be heterogeneous, referred to as distribution heterogeneity. Furthermore, the associations between features and responses could vary due to distinct scientific goals or other factors, which we refer to as posterior heterogeneity. In the following, we provide concrete motivating examples to illustrate these challenges in different problems.

**Example 1: Medical multi-source datasets.** Multi-source data are widely observed in medical applications and offer more comprehensive information than single-source data. For example, the Alzheimer's Disease Neuroimaging Initiative (ADNI) dataset includes medical imaging, biosamples, gene expression, and demographic information (Mueller et al., 2005a;b). However, entire blocks of data are often missing when certain sources become unnecessary or infeasible to collect due to known factors or patient conditions (Madden et al., 2016).

**Example 2: Single-cell multi-omics datasets.** Data from different experimental batches often exhibit distribution heterogeneity across various omics measurements. For instance, transcriptome data collected from different batches can display varying patterns due to differences in experimental conditions or technical variability (Cao et al., 2022a;b). In multi-omics datasets, sequencing data distributions also differ across various cancer types (Subramanian et al., 2020).

**Example 3: Combining randomized controlled trials (RCTs) and observational data.** Combining RCTs and observational data has become effective for deriving causal effects due to the high costs and limited participant numbers in RCTs (Colnet et al., 2024). However, RCTs and observational data often exhibit posterior heterogeneity (Li et al., 2024a); for instance, causal effects in RCTs may differ from associations in observational data due to the controlled conditions of RCTs (Imbens & Rubin, 2015).

**Challenges.** The challenge in MTL is to incorporate various forms of heterogeneity, each introducing a unique challenge. Block-wise heterogeneity complicates the integration of data as missing patterns vary across tasks, making it difficult to leverage shared information efficiently. For example, in the ADNI dataset, imaging features are present in all datasets, but genetic information is available only in specific subsets (Xue & Qu, 2021). In addition, distribution heterogeneity can also lead to biased or misleading scientific conclusions if not addressed properly. For instance, in multi-omics datasets, sequencing data vary significantly across different cancer types (Subramanian et al., 2020). Lastly, posterior heterogeneity affects the accuracy of predictions. For example, the relationships identified in RCTs often do not align with those observational data collected in real-life settings (Kent et al., 2018; 2020). While each type of heterogeneity imposes its own challenge, addressing all three challenges simultaneously under a unified framework presents significant obstacles, and to our best knowledge, current MTL methods are not equipped to handle these intricate dilemmas.

**Contributions.** In this work, we propose a unified MTL framework to address three types of heterogeneity in MTL. There are three key contributions: First, we propose a novel block-wise missing imputation method which effectively handles distribution heterogeneity by learning both shared and task-specific representations, uncovering complex structures between sources, and enabling better generalization during imputation. Second, we disentangle the associations between all input features and responses into shared and task-specific components, allowing for the effective integration of information while adapting to differences across tasks. Third, we propose an MTL architecture consisting of two parts to construct these associations. The first part builds heterogeneous feature spaces, while the second part learns responses, jointly addressing both distribution and posterior heterogeneity. We validate the proposed framework on synthetic and real-world datasets, demonstrating its superior performance in handling block-wise missing data and various levels of heterogeneity.

## 2 RELATED WORK

**Multi-source data integration.** Several related works on multi-source data collected for the same set of samples fall within the Joint and Individual Variation Explained (JIVE) framework. These methods are classified as unsupervised or supervised JIVE, depending on the presence of responses. Unsupervised JIVE and its variants learn joint, individual, and partially shared structures from multiple data matrices through low-rank approximations (Lock et al., 2013; Feng et al., 2018; Gaynanova & Li, 2019; Choi & Jung, 2022; Yi et al., 2023; James et al., 2024). Supervised JIVE, on the other hand, focuses on regression for multi-source data (Gao et al., 2021; Palzer et al., 2022; Zhang & Gaynanova, 2022; Wang & Lock, 2024). Similarly, factor models have been applied to multi-source data in a supervised setting (Shu et al., 2020; Li & Li, 2022; Anceschi et al., 2024). While these methods can effectively address distribution heterogeneity across different sources in linear settings, they are limited in scope, as they capture only simple data structures within a single task.

**Multi-source block-wise missing data integration.** Recently, several methods have been developed to address block-wise missing data. These methods can be divided into two categories based on whether imputation is involved. Imputation-based methods assume consistent correlations between different sources across datasets, allowing for the imputation of missing blocks (Gao & Lee, 2017; Le Morvan et al., 2021; Xue & Qu, 2021; Xue et al., 2021; Zhou et al., 2021; Ouyang et al., 2024). For example, Xue & Qu (2021) and Xue et al. (2021) construct estimating equations using all available information and integrate informative estimating functions to achieve efficient estimators. On the other hand, non-imputation-based methods focus on learning the covariance matrices among predictors and between the response and predictors from the observed blocks (Yuan et al., 2012; Xiang et al., 2014; Yu et al., 2020; Li et al., 2024b). While these methods perform well in the absence of distribution shift and posterior shift, effectively utilizing all block-wise missing data, they struggle to handle distribution or posterior heterogeneity.

**Multi-task learning (MTL).** There is a growing literature on learning multiple tasks simultaneously with a shared model; see Zhang & Yang (2018); Crawshaw (2020); Zhang & Yang (2021) for reviews. Here, we primarily focus on MTL with deep neural networks, as these networks can capture more complex relationships. These methods can be broadly classified into four categories: The first category is balancing individual loss functions for different tasks, which is a common approach to ease multi-task optimization (Du et al., 2018; Gong et al., 2019; Hang et al., 2023; Wu et al., 2024). The second category involves regularization, especially in the form of hard parameter sharing (Subramanian et al., 2018; Liu et al., 2019; Maziarka et al., 2022) and soft parameter sharing (Ullrich et al., 2017; Lee et al., 2018; Han et al., 2024). The third category addresses the challenge of negative transfer, where explicit gradient modulation is used to alleviate conflicts in learning dynamics between tasks (Lopez-Paz & Ranzato, 2017; Chaudhry et al., 2018; Maninis et al., 2019; Abdollahzadeh et al., 2021; Hu et al., 2022; Wang et al., 2024b). The fourth category uses knowledge distillation to transfer knowledge from single-task networks to a multi-task student network (Rusu et al., 2015; Teh et al., 2017; Clark et al., 2019; D'Eramo et al., 2024). Although MTL can integrate data from multiple tasks, it is limited in addressing different types of heterogeneity and is constrained by the assumption of a fully observed setting.

Most related work focuses on addressing a single challenge, such as posterior heterogeneity or the missing data problem, but typically fails to address all challenges simultaneously. In contrast, our proposed method extends these approaches by tackling both distribution and posterior heterogeneity in a block-wise missing setting. This enables a more comprehensive integration of data across tasks, resulting in improved performance in MTL.

# 3 Two-step MTL for heterogeneous multi-source block-wise missing data

**Notation.** We introduce the notations used in this paper. Vectors and matrices are denoted by $\boldsymbol{x}$ and $\boldsymbol{X}$, respectively. The $\ell_1$ and $\ell_2$ norms of a vector $\boldsymbol{x}$ are $\|\boldsymbol{x}\|_1$ and $\|\boldsymbol{x}\|_2$, and the Frobenius norm of a matrix $\boldsymbol{X}$ is $\|\boldsymbol{X}\|_F$. The symbol $|$ represents concatenation. For example, $[\boldsymbol{x}_1|\boldsymbol{x}_2]$ denotes concatenating a $p_1 \times 1$ vector $\boldsymbol{x}_1$ and a $p_2 \times 1$ vector $\boldsymbol{x}_2$ into a $(p_1 + p_2) \times 1$ vector. Similarly, $[\boldsymbol{X}_1|\boldsymbol{X}_2]$ denotes concatenating an $n \times p_1$ matrix $\boldsymbol{X}_1$ and an $n \times p_2$ matrix $\boldsymbol{X}_2$ into an $n \times (p_1 + p_2)$ matrix. We define $[r] = \{1, 2, \ldots, r\}$ as the set of integers from $1$ to $r$.

**Problem Description.** Suppose we have data from $T$ tasks, with features collected from $T + 1$ sources. For all tasks, we assume that a common source, called the anchoring source, is observed. Additionally, each task has its own task-specific source, denoted as $\boldsymbol{x}_s^t$ for the $s$-th source in the $t$-th task. Specifically, $\boldsymbol{x}_0^t$ represents the anchoring source observed in the $t$-th task, and $\boldsymbol{x}_t^t$ denotes the task-specific source for the $t$-th task, while $\{\boldsymbol{x}_s^t\}_{s \neq 0, t}$ are missing. For the $t$-th task, we observe $n_t$ samples $\{[\boldsymbol{x}_{0,i}^t | \boldsymbol{x}_{t,i}^t], y_i^t\}_{i=1}^{n_t}$. This block-wise missing pattern is common in real-world applications. For example, in biomedical data, some measurements (data sources) are widely observed for all subjects, while some measurements are only collected to a subgroup

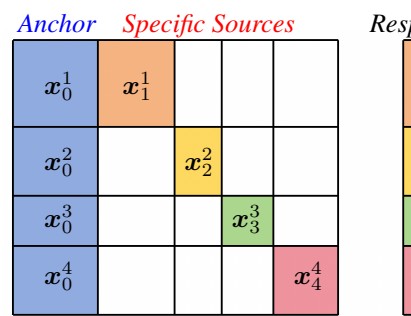

Figure 1: Block-wise missing pattern for $4$ tasks and $5$ sources, including an anchoring source and task-specific sources.

of subjects due to various reasons. Concretely, in the ADNI data that we analyzed in Section 4.3, MRI is crucial to monitor the cognitive impairment development of Alzheimer's patients, so it is measured for all subjects, while gene expression and PET images are less crucial and are only observed for two subgroups separately. Another example is the split questionnaire design, which aims to reduce respondent fatigue and improve response rates by assigning different subsets of the questionnaire to different sampled respondents (Lin et al., 2023). In Figure 1, we provide an example of a block-wise missing pattern for $4$ tasks and $5$ sources, where the blue source $\boldsymbol{x}_0^t$ for $t \in [4]$ represents the anchoring source observed by all four tasks, and each task also has a uniquely observed specific source $\boldsymbol{x}_t^t$ for $t \in [4]$. Our goal is to perform MTL on these tasks with block-wise missing data.

Figure 1 illustrates one of the challenges in MTL. Each task has different missing blocks; for example, in the first task, sources 2, 3, and 4 are missing, while in the second task, sources 1, 3, and 4 are missing. Furthermore, both distribution and posterior heterogeneity across tasks complicate the application of standard imputation methods (Nair et al., 2019; He et al., 2024a;b) and MTL methods (Kouw & Loog, 2018; Lee et al., 2024; Maity et al., 2024).

### 3.1 HETEROGENEOUS BLOCK-WISE IMPUTATION

In this section, we propose the first step, Heterogeneous Block-wise Imputation (HBI) for imputing the missing blocks while leveraging distribution heterogeneity across tasks. HBI extracts disentangled hidden representations from the anchoring source $x_0$, including a shared representation across tasks and a task-specific representation for each task. The shared representation is then used to impute the missing blocks, improving generalization across tasks.

For $T$ tasks and $T+1$ sources, we impute the task-specific sources in a parallel fashion. For each task-specific source $s \neq 0$, we utilize the anchoring source across all tasks and $x_s^s$ to impute the unobserved blocks $\{x_s^t\}_{t \neq s}$. In particular, for the $t$-th source, only the $t$-th task has observed values for the features $x_t^t$. The imputation aims to use the observed $x_0^t$ and $x_t^t$ along with $x_0^{-t} = \{x_0^r\}_{r \neq t}$ to estimate the missing features in the $t$-th source for the other $T-1$ tasks, where $x_t^{-t} = \{x_t^r\}_{r \neq t}$ are unobserved.



Figure 2: Illustration of parallel imputation for task-specific sources.

For example, in Figure 2, we use information from $x_0^1$, $x_1^1$, and $x_0^{-1} = \{x_0^2, x_0^3, x_0^4\}$ to impute the missing blocks $x_1^{-1} = \{x_1^2, x_1^3, x_1^4\}$ for the task 1-specific source.

This is accomplished by learning a model that exploits both the shared and task-specific information of the data, allowing for accurate prediction of missing values based on the available observed data. To fully integrate multi-source information, we leverage an encoder-decoder framework, which is well-suited for capturing non-linear relationships in data. Let $E_c(\cdot)$ be a common encoder that maps $\{x_0^t, x_0^{-t}\}$ to shared representations $f_c^t = E_c(x_0^t)$ and $f_c^{-t} = E_c(x_0^{-t})$ across all $T$ tasks. Let $E_p^t(\cdot)$ and $E_p^{-t}(\cdot)$ be task-specific encoders that map $x_0^t$ and $x_0^{-t}$ to task-specific representations $g^t = E_p^t(x_0^t)$ and $g^{-t} = E_p^{-t}(x_0^{-t})$. Then, $D(f, g)$ serves as a decoder that reconstructs the anchoring source $x_0$ from $f$ and $g$. Finally, $G(f)$ is a predictor that maps the shared representation $f$ to the task $t$-specific source $x_t$. The resulting heterogeneous block-wise imputation model is illustrated in Figure 3.

In Figure 3, we assume that the relationship between the anchoring source $x_0$ and the task $t$-specific source $x_t$ can be borrowed through the shared representations $f$, the common encoder $E_c(\cdot)$, and the decoder $G(\cdot)$. This allows us to utilize the shared information (reflected in $f^t$ and $f^{-t}$) for imputation, while also accounting for the heterogeneity between $x_0^t$ and $x_0^{-t}$ (reflected in $g^t$ and $g^{-t}$). Existing imputation methods often learn the relationship between $x_0$ and $x_t$ within the $t$-th task and apply the rela-

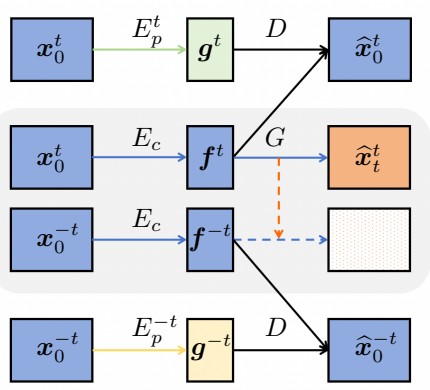

Figure 3: Illustration of HBI for the task $t$-specific source $x_t$. A common encoder $E_c(\cdot)$ learns to capture representation components that are shared among tasks. Task-specific encoders $E_p(\cdot)$ (one for the $t$-th task, and one for the other $T-1$ tasks) learn to capture task-specific components of the representations. A decoder learns to reconstruct the anchoring source $x_0$ by using both shared and task-specific representations. The shared part of the relationship between the anchoring source $x_0$ and the task $t$-specific source $x_t$ can be borrowed through $E_c(\cdot)$ and $G(\cdot)$ for imputation. See the text for more information.

tionship to other tasks, overlooking distribution heterogeneity (Xue et al., 2021; Zhou et al., 2021). Moreover, common imputation methods rely on parametric models which fail to capture complex re-

lationships in missing data (Xue & Qu, 2021; Li et al., 2023). However, our HBI method effectively overcomes these obstacles. Notably, HBI's extraction of the common components in the relationships between sources across different tasks shares similarities with domain adaptation (Mansour et al., 2008; Bousmalis et al., 2016; Tzeng et al., 2017; Farahani et al., 2021) but focuses on completing block-wise missing data. This architecture effectively models complex data structures and interactions, providing a robust tool for understanding intricate patterns. The resulting optimization can be formulated as:

$$(\widehat{E}_c(\cdot), \widehat{E}_p^t(\cdot), \widehat{E}_p^{-t}(\cdot), \widehat{D}(\cdot), \widehat{G}(\cdot)) = \arg\min\{\mathcal{L}_{\text{pre}} + \mathcal{L}_{\text{recon}}\}, \tag{1}$$

In (1), the prediction loss $\mathcal{L}_{\text{pre}}$ trains the model to predict $\boldsymbol{x}_t^t$, the target of interest, which is applied only to the $t$-th task. We use the following loss function:

$$\mathcal{L}_{\text{pre}} = \sum_{i=1}^{n_t} l(\boldsymbol{x}_{t,i}^t, G(E_c(\boldsymbol{x}_{0,i}^t))),$$

where $\boldsymbol{x}_i$ denotes the observed sample, and $l(\cdot, \cdot)$ can be the mean squared error for continuous outcomes or cross-entropy for binary outcomes (this applies similarly to the following symbols). For the reconstruction loss in (1),

$$\mathcal{L}_{\text{recon}} = \sum_{i=1}^{n_t} l(\boldsymbol{x}_{0,i}^t, D(E_c(\boldsymbol{x}_{0,i}^t), E_p^t(\boldsymbol{x}_{0,i}^t))) + \sum_{i=1}^{n_{-t}} l(\boldsymbol{x}_{0,i}^{-r}, D(E_c(\boldsymbol{x}_{0,i}^{-t}), E_p^{-t}(\boldsymbol{x}_{0,i}^{-t}))),$$

where $n_{-t} = \sum_{r \neq t} n_r$. Then, we can train (1) to obtain the estimators $\widehat{E}_c(\cdot)$ and $\widehat{G}(\cdot)$. Consequently, we compute $\widehat{\boldsymbol{x}}_t^{-t} = \widehat{G}(\widehat{E}_c(\boldsymbol{x}_0^{-t}))$. Note that (1) is constructed based on task $t$-specific source imputation. Similarly, we can construct imputations for the other $T-1$ sources. When performing imputation for different sources using HBI, the learned hidden representations and corresponding generative functions adapt dynamically. This adaptation is crucial as it allows the model to accommodate the unique information of each source. The complete algorithm for parallel heterogeneous imputation is provided in Appendix A.4.

Our proposed HBI method ensures that the imputation model leverages common information across tasks while incorporating the heterogeneity of each task. By decomposing the latent space into shared and task-specific components, we gain a nuanced understanding of how input features from different sources interact, thereby enhancing imputation accuracy.

## 3.2 HETEROGENEOUS MULTI-TASK LEARNING

In this section, we propose our MTL framework to accommodate distribution and posterior heterogeneity given the imputed blocks from HBI. Similar to the disentangled representations for features, we also model the association between features and responses as two components: a shared function mapping and a task-specific function mapping. Specifically, for the $t$-th task, we assume that the relationship between the response $y^t$ and the features $[\boldsymbol{x}_0^t | \boldsymbol{x}_1^t | \cdots | \boldsymbol{x}_T^t]$ is given by:

$$y^t = \psi_c([\boldsymbol{x}_0^t | \boldsymbol{x}_1^t | \cdots | \boldsymbol{x}_T^t]) + \psi_p^t([\boldsymbol{x}_0^t | \boldsymbol{x}_1^t | \cdots | \boldsymbol{x}_T^t]), \tag{2}$$

where $[\boldsymbol{x}_0^t | \boldsymbol{x}_1^t | \cdots | \boldsymbol{x}_T^t]$ influence $y^t$ through a shared mapping $\psi_c(\cdot)$ and a task-specific mapping $\psi_p^t(\cdot)$. Equation 2 extends traditional meta-analysis, which often assumes a linear relationship in the $t$-th task as $y^t = [\boldsymbol{x}_0^t | \boldsymbol{x}_1^t | \cdots | \boldsymbol{x}_T^t]^\top \boldsymbol{\beta}^t + \varepsilon$, where $\boldsymbol{\beta}^t$ includes a common component $\boldsymbol{\mu}$ shared across all $T$ tasks and a unique component $\boldsymbol{\alpha}^t$ for each task (Chen et al., 2021; Cai et al., 2022; Maity et al., 2022). Traditional meta-analysis is incapable of accommodating non-linear relationships or varying effects. In contrast, we propose a flexible framework which accommodates non-linearities and integrates task-specific information.

To construct the shared mapping $\psi_c(\cdot)$ and the task-specific mappings $\{\psi_p^t\}_{i=1}^T$ jointly, we consider an MTL architecture comprising two parts. The first part builds heterogeneous feature spaces, while the second part learns responses for all $T$ tasks. Specifically, for the $t$-th task, following HBI in Section 3.1, we obtain samples with reconstructed features $\{(\boldsymbol{x}_{0,i}^t, \ldots, \widehat{\boldsymbol{x}}_{t-1,i}^t, \boldsymbol{x}_{t,i}^t, \widehat{\boldsymbol{x}}_{t+1,i}^t, \ldots, \widehat{\boldsymbol{x}}_{T,i}^t), y_i^t\}_{i=1}^{n_t}$. These features can then be integrated to capture both shared and task-specific representations, enabling the utilization of the combined data while addressing task-specific heterogeneity. During HBI, the components $\{\widehat{\boldsymbol{x}}_s^t\}_{s \neq 0, t}$ are primarily predicted

using the anchoring source $\boldsymbol{x}_0^t$, indicating that $\boldsymbol{x}_0^t$ serves as a common basis. To prevent redundancy, we extract shared representations solely from the anchoring source $\boldsymbol{x}_0^t$. Specifically, we define

$$\boldsymbol{h}^t = \phi_c(\boldsymbol{x}_0^t), \tag{3}$$

where $\phi_c(\cdot)$ is a shared encoder that learns hidden information from the anchoring source for all tasks. Meanwhile, task heterogeneity is captured by extracting representations from all features, creating a framework in which shared representations provide a common foundation, while task-specific details can still be preserved. For the $t$-th task, we define:

$$\boldsymbol{k}^t = \phi_p^t([\boldsymbol{x}_0^t|\cdots|\widehat{\boldsymbol{x}}_{t-1}^t|\boldsymbol{x}_t^t|\widehat{\boldsymbol{x}}_{t+1}^t|\cdots|\widehat{\boldsymbol{x}}_T^t]), \tag{4}$$

where $\phi_p^t$ is a task-specific encoder which maps the unique information within the $t$-th task. In (3) and (4), the heterogeneous feature spaces are fully captured using all data information. The task-specific representations $\boldsymbol{k}$ capture complex interactions between different sources unique to each task, aided by HBI in Section 3.1. In practice, such interactions are crucial. For example, in the ADNI dataset, there are intricate relationships between images and gene expression. Equation 4 accounts for this heterogeneous information. However, previous work (Moon & Carbonell, 2017; Bica & van der Schaar, 2022) often oversimplifies these interactions by focusing only on task-specific sources, neglecting a wealth of shared information from other tasks.

For the second part, we consider a network architecture for learning responses in all $T$ tasks, consisting of $L$ layers with both shared and task-specific subspaces (Ruder et al., 2019; Curth & Van der Schaar, 2021; Bica & van der Schaar, 2022). For simplicity, in the $t$-th task, let $\bar{\boldsymbol{k}}_l^t$ and $\bar{\boldsymbol{h}}_l^t$ represent the inputs, and $\boldsymbol{k}_l^t$ and $\boldsymbol{h}_l^t$ the outputs of the $l$-th layer. For $l = 1$, set $\bar{\boldsymbol{k}}_1^t = [\boldsymbol{h}^t|\boldsymbol{k}^t]$ and $\bar{\boldsymbol{h}}_1^t = \bar{\boldsymbol{k}}_1^t$. For $l > 1$, the inputs to the $(l+1)$-th layer are given by $\bar{\boldsymbol{k}}_{l+1}^t = [\boldsymbol{h}_l^t|\boldsymbol{k}_l^t]$ and $\bar{\boldsymbol{h}}_l^t = [\boldsymbol{h}_l^t]$. Let $g^t(\cdot)$ be the association function in the $t$-th task, defined as $g^t([\boldsymbol{h}_L^t|\boldsymbol{k}_L^t]) = \psi_c([\boldsymbol{x}_0^t|\boldsymbol{x}_1^t|\cdots|\boldsymbol{x}_T^t]) + \psi_p^t([\boldsymbol{x}_0^t|\boldsymbol{x}_1^t|\cdots|\boldsymbol{x}_T^t])$, where $g^t(\cdot)$ is a linear function for continuous outcomes and a sigmoid function for binary ones.

Figure 4 illustrates the construction of the shared mapping $\psi_c(\cdot)$ and the task-specific mappings $\psi_p^1(\cdot)$ and $\psi_p^2(\cdot)$ for two tasks. For task 1, the input features consist of $[\boldsymbol{x}_0^1|\boldsymbol{x}_1^1|\widehat{\boldsymbol{x}}_2^1]$, where $\widehat{\boldsymbol{x}}_2^1$ represents the imputed source. In the first part, we build the heterogeneous feature space by extracting the shared representation $\boldsymbol{h}^1 = \phi_c(\boldsymbol{x}_0^1)$ and the task-specific representation $\boldsymbol{k}^1 = \phi_p^1([\boldsymbol{x}_0^1|\boldsymbol{x}_1^1|\widehat{\boldsymbol{x}}_2^1])$. Similarly, for task 2, we extract $\boldsymbol{h}^2 = \phi_c(\boldsymbol{x}_0^2)$ and $\boldsymbol{k}^2 = \phi_p^2([\boldsymbol{x}_0^2|\widehat{\boldsymbol{x}}_1^2|\boldsymbol{x}_2^2])$. Next, we utilize the pairs $\{\boldsymbol{h}^1, \boldsymbol{k}^1\}$ and $\{\boldsymbol{h}^2, \boldsymbol{k}^2\}$ to model the responses $y^1$ and $y^2$, respectively. In Figure 4, the blue mapping illustrates the shared mapping $\psi_c(\cdot)$, while the orange and yellow mappings represent the task-specific mappings $\psi_p^1(\cdot)$ and $\psi_p^2(\cdot)$ for tasks 1 and 2, respectively.

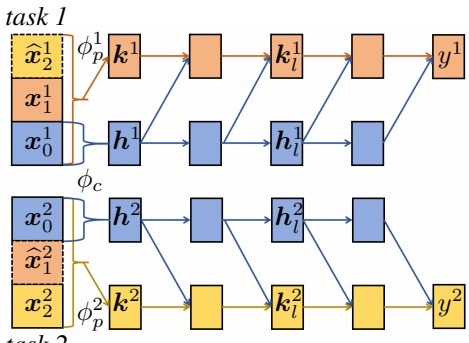

Figure 4: Illustration of the construction of shared mapping and task-specific mappings for two tasks.

The above construction allows us to define the following integrated loss across all $T$ tasks:

$$\mathcal{L}_{\text{integ}} = \sum_{t=1}^{T} \sum_{i=1}^{n_t} l(y_i^t, g^t([\boldsymbol{h}_{L,i}^t|\boldsymbol{k}_{L,i}^t])). \tag{5}$$

Similar to Bousmalis et al. (2016), we also incorporate an orthogonality regularizer, defined as:

$$\mathcal{R}_{\text{orth}} = \sum_{t=1}^{T} \|(\boldsymbol{H}^t)^\top \boldsymbol{K}^t\|_F^2, \tag{6}$$

where $\boldsymbol{H}^t$ and $\boldsymbol{K}^t$ are matrices whose rows are the hidden representations $\boldsymbol{h}^t$ and $\boldsymbol{k}^t$, respectively. Furthermore, in (4), the input is $[\boldsymbol{x}_0^t|\cdots|\widehat{\boldsymbol{x}}_{t-1}^t|\boldsymbol{x}_t^t|\widehat{\boldsymbol{x}}_{t+1}^t|\cdots|\widehat{\boldsymbol{x}}_T^t]$, where $\boldsymbol{x}_0^t$ and $\boldsymbol{x}_t^t$ are the observed data, and $\{\widehat{\boldsymbol{x}}_s^t\}_{s\neq 0,t}$ are obtained through imputation. Since imputation can introduce errors, we also downweight the imputed data $\{\widehat{\boldsymbol{x}}_s^t\}_{s\neq 0,t}$ compared to observed data for learning $\boldsymbol{k}^t$ by

applying a regularizer to the parameters of the first layer of the encoder $\phi_p^t(\cdot)$, defined as:

$$\mathcal{R}_{\text{imp}} = \sum_{t=1}^{T} \sum_{s \neq 0, t} \|\boldsymbol{\Theta}_{s,p,1}^t\|_F^2, \tag{7}$$

where $\boldsymbol{\Theta}_{s,p,1}^t$ are the parameters of the first layer of $\phi_p^t(\cdot)$ corresponding to $\{\widehat{\boldsymbol{x}}_s^t\}_{s \neq 0,t}$. This regularizer downweights potentially less accurate imputed features by penalizing the magnitude of the encoder parameters, fostering a model more robust to imputation errors. To further reduce redundancy between the shared and task-specific layers, we introduce an orthogonal regularizer (Ruder et al., 2019; Bica & van der Schaar, 2022). Let $d_{c,l-1}^t$ and $d_{p,l-1}^t$ be the dimensions of $\boldsymbol{h}_{l-1}^t$ and $\boldsymbol{k}_{l-1}^t$, the outputs of the $(l-1)$-th layer. Denote the weights in the $l$-th layer as $\boldsymbol{\Theta}_{c,l}^t \in \mathbb{R}^{d_{c,l-1}^t \times d_{c,l}^t}$ and $\boldsymbol{\Theta}_{p,l}^t \in \mathbb{R}^{(d_{c,l-1}^t + d_{p,l-1}^t) \times d_{p,l}^t}$. We apply the following regularizer:

$$\mathcal{R}_{\text{dr}} = \sum_{t=1}^{T} \sum_{l=1}^{L} \|(\boldsymbol{\Theta}_{c,l}^t)^\top \boldsymbol{\Theta}_{p,l,1:d_{c,l-1}^t}^t\|_F^2. \tag{8}$$

By combining the losses from (5), (6), (7), and (8), we train the set of all parameters $\boldsymbol{\Theta}$ using the following integrated loss function:

$$(\widehat{\psi}_c, \{\widehat{\psi}_p^t\}_{1 \leq t \leq T}) = \arg\min_{\boldsymbol{\Theta}} \{\mathcal{L}_{\text{integ}} + \gamma \mathcal{R}_{\text{orth}} + \delta \mathcal{R}_{\text{imp}} + \kappa \mathcal{R}_{\text{dr}}\},$$

where $\gamma$, $\delta$, and $\kappa$ are weights controlling the balance among different terms. The more detailed algorithm for heterogeneous MTL is provided in Appendix A.4.

## 4 EXPERIMENTS

In this section, we conduct extensive numerical experiments, including two-task MTL, multi-task MTL with more than two tasks, and an application to the ADNI real dataset. The numerical experiments demonstrate that our proposed two-step MTL method effectively aggregates information in the presence of block-wise, distribution, and posterior heterogeneity.

### 4.1 MTL FOR TWO TASKS

We address a common real-world scenario involving MTL with two tasks for illustration. The data generation process (DGP) is as follows:

**DGP**: For Task 1, the features are denoted as $\boldsymbol{x}^1 = [\boldsymbol{x}_0^1|\boldsymbol{x}_1^1|\boldsymbol{x}_2^1]$ and follow a Gaussian distribution with mean $\boldsymbol{0}$ and an exchangeable covariance matrix. The variance is fixed at 1, and the covariance structure is specified as $(\rho_1)^{0.01|i-j|}$. We randomly generate $n_1$ samples, with the third block $\boldsymbol{x}_2^1$ missing in Task 1. For Task 2, the features are denoted as $\boldsymbol{x}^2 = [\boldsymbol{x}_0^2|\boldsymbol{x}_1^2|\boldsymbol{x}_2^2]$, where $\boldsymbol{x}^2$ follows a Gaussian distribution with mean $\boldsymbol{0}$, variance 1, and covariance structure $(\rho_2)^{0.01|i-j|}$. In this task, we generate $n_2$ samples, with the second block $\boldsymbol{x}_1^2$ missing. The responses are defined as:

$$y^1 = \alpha \sum_{d=1}^{p} v_{c,d}(x_d^1)^2/p + (1-\alpha) \sum_{d=1}^{p} v_{1,d} x_d^1/p + \varepsilon_1,$$

$$y^2 = \alpha \sum_{d=1}^{p} v_{c,d}(x_d^2)^2/p + (1-\alpha) \sum_{d=1}^{p} v_{2,d} x_d^2/p + \varepsilon_2,$$

where $p = \sum_{s=0}^{2} p_s$, with $p_s$ being the dimension of the $s$-th source, and the subscript $d$ denotes the $d$-th element of a vector (this notation applies to subsequent symbols as well). The parameters $v_c$, $v_1$, and $v_2$ are sampled from $N(-10, 10^2)$, and the noise terms $\varepsilon_1 \sim N(0, \sigma_1^2)$ and $\varepsilon_2 \sim N(0, \sigma_2^2)$. The parameter $\alpha$ controls the level of sharing across tasks. Additionally, our DGP accounts for nonlinear relationships by element-wise square, further increasing the complexity of MTL. For evaluation, we calculate the average root-mean-squared error (RMSE) on the testing data, as defined in Appendix A.2. We conduct experiments under various settings to compare the proposed MTL for heterogeneous multi-source block-wise missing data (MTL-HMB) against existing methods, including

Single Task Learning (STL) and Transfer Learning for Heterogeneous Data (HTL) (Bica & van der Schaar, 2022).

**Setting A: Effect of covariance parameters.** We set $n_1 = n_2 = 300$, $p_0 = 100$, $p_1 = p_2 = 25$, $\alpha = 0.3$, and $\sigma_1 = \sigma_2 = 0.1$. To examine the impact of the covariance parameters $\rho_1$ and $\rho_2$, we set $\rho_1 = \rho_2$ and vary them from $0.5$ to $0.95$, assuming no distribution heterogeneity across datasets. As shown in Figure 5(a), increasing correlation improves prediction accuracy across all methods. The proposed MTL-HMB is the best performer. Specifically, at $\rho = 0.95$, it outperforms the others by more than $28.33\%$. Even at $\rho = 0.5$, despite imputation errors, our method maintains an advantage. This demonstrates that imputation enhances prediction, especially when distribution heterogeneity is absent.

**Setting B: Effect of heterogeneous covariance parameters.** We set $n_1 = n_2 = 300$, $p_0 = 100$, $p_1 = p_2 = 25$, $\alpha = 0.3$, and $\sigma_1 = \sigma_2 = 0.1$. To assess the impact of heterogeneous covariance, we fix $\rho_1 = 0.95$ and vary $\rho_2$ from $0.5$ to $0.9$. Smaller $\rho_2$ indicates greater heterogeneity and weaker correlations in Task 2, making predictions more challenging. Figure 5(b) shows that as $\rho_2$ increases, all methods improve, and our approach consistently leads. At the highest level of heterogeneity, MTL-HMB outperforms HTL by over $20.91\%$. Moreover, HTL shows no advantage over STL, indicating that transfer learning struggles with distribution heterogeneity. In contrast, the proposed method effectively solves the heterogeneity challenge through imputation, achieving better predictive accuracy.

**Setting C: Effect of heterogeneous mappings.** We set $n_1 = n_2 = 300$, $p_0 = 100$, $p_1 = p_2 = 25$, $\rho_1 = \rho_2 = 0.8$, and $\sigma_1 = \sigma_2 = 0.1$. The parameter $\alpha$ is varied to control the level of information sharing in the mappings to the response. A larger $\alpha$ indicates more shared information. With $\rho_1 = \rho_2$ fixed, heterogeneity is governed solely by $\alpha$. Figure 5(c) shows that as $\alpha$ increases, the magnitude of $y$ also increases, resulting in higher average RMSEs. Except at $\alpha = 0.1$, HTL consistently outperforms STL, indicating its advantage in incorporating posterior shift. Overall, the proposed MTL-HMB performs the best across all settings, even in the absence of distribution heterogeneity.

**Setting D: Effect of sample sizes.** We set $p_0 = 100$, $p_1 = p_2 = 25$, $\rho_1 = 0.95$, $\rho_2 = 0.7$, $\alpha = 0.3$, and $\sigma_1 = \sigma_2 = 0.1$. The sample sizes $n_1$ and $n_2$ vary as $n_1 = n_2 = k \times 100$ for $k = 1, \ldots, 6$. Figure 5(d) shows that as the sample size increases, average RMSEs decrease, and the corresponding variability of estimator is reduced across all methods. Our method consistently performs best, with an improvement of over $18.13\%$ compared to HTL. This is particularly notable at smaller sample sizes such as $100$, where MTL-HMB outperforms HTL and STL by $37.13\%$ and $38.06\%$, respectively. Additionally, HTL does not significantly outperform STL, indicating difficulty in handling distribution heterogeneity.

**Setting E: Effect of dimensions.** We set $n_1 = n_2 = 300$, $\rho_1 = 0.95$, $\rho_2 = 0.7$, $\alpha = 0.3$, and $\sigma_1 = \sigma_2 = 0.1$. To assess the impact of dimensions $p_1$, $p_2$, and $p_3$, we fix $p_1 = 100$ and vary $p_2 = p_3 = k \times 25$ for $k = 1, \ldots, 4$. Figure 5(e) shows that increasing dimensions make prediction more challenging, leading to higher average RMSEs for all methods. Our method consistently outperforms the others, with at least $7.88\%$ and $10.73\%$ improvements over HTL and STL, respectively. Moreover, it exhibits greater stability, reflected by lower RMSEs at both the 75th and 25th percentiles.

**Setting F: Effect of heterogeneous noise levels.** We set $n_1 = n_2 = 300$, $p_0 = 100$, $p_1 = p_2 = 25$, $\rho_1 = 0.95$, $\rho_2 = 0.7$. By fixing $\sigma_2 = 0.1$ and varying $\sigma_1$ from $0.1$ to $0.5$, we assess the impact of different noise levels. Figure 5(f) shows that HTL lacks a clear advantage over STL, indicating that distribution heterogeneity leads to degenerated HTL's performance. Our MTL-HMB consistently outperforms the competing methods, demonstrating robustness in addressing both distribution and posterior heterogeneity. Furthermore, MTL-HMB excels at lower prediction levels, with neither STL nor HTL matching its performance at the 25th percentile.

## 4.2 MTL FOR MULTIPLE TASKS

To save space, the DGP for number of tasks greater than 2 is detailed in Appendix A.2. We select $T = 2$, $T = 3$, and $T = 4$, with the prediction results summarized in Figure 6. As shown in Figure 6(a), increasing the number of heterogeneous tasks makes prediction more challenging, resulting in higher average RMSEs. This underscores the complexity of integrating diverse data. Nevertheless,

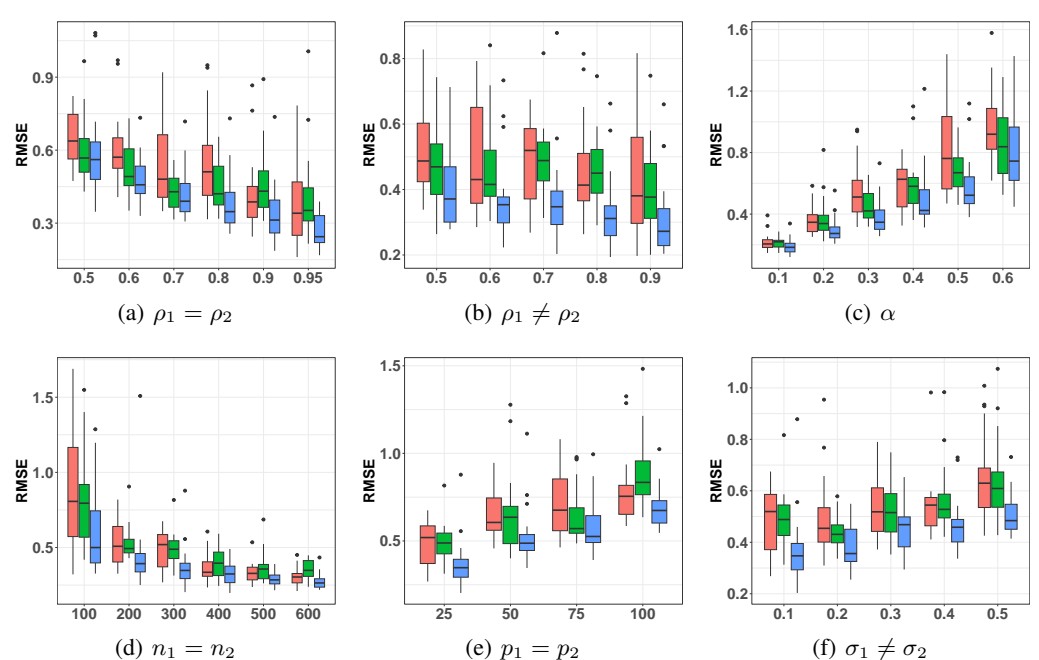

Figure 5: Boxplots of average RMSEs under **Settings A** to **F** for the three methods. The methods are distinguished by color: **orange** for STL, **green** for HTL, and **blue** for the proposed MTL-HMB.

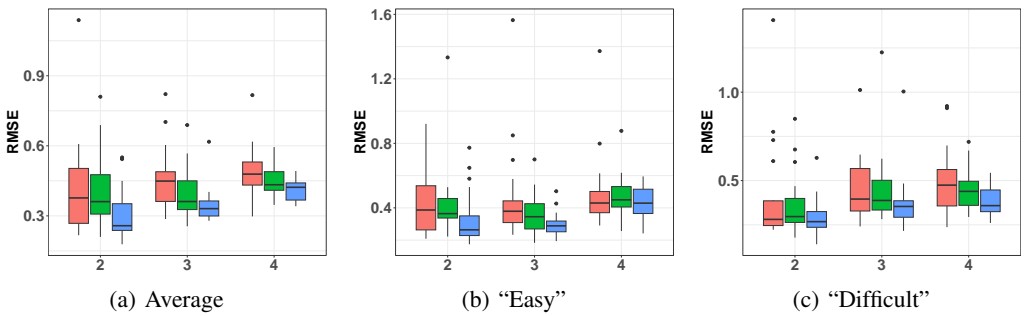

Figure 6: Boxplots of average RMSEs under multiple settings across three methods. The methods are differentiated by color: **orange** for STL, **green** for HTL, and **blue** for our proposed MTL-HMB.

our MTL-HMB consistently outperforms the other methods and shows the smallest RMSE standard deviation, indicating greater robustness and reliability. We focus on the first task, considering it the "easiest" due to the highest observed correlations. Figure 6(b) shows that for this "easy" task, as more tasks are integrated, the improvement of our method decreases due to increasing heterogeneity. Additionally, we observe that in two-task learning, the second task is the most challenging; in three-task learning, it is the third task; and in four-task learning, it is the fourth task. This pattern indicates that as the number of integrated tasks increases, the complexity of learning escalates, particularly for the most recently added task. To quantify these challenges, we compile the RMSEs for the most difficult tasks in Figure 6(c), which shows that our MTL-HMB excels in these challenging tasks, consistently outperforming the other methods. For example, in the four-task integration, our method achieves over $18.22\%$ improvement compared to the HTL.

## 4.3 ADNI REAL DATA APPLICATION

We perform MTL using the ADNI database. The first task has 72 samples with features from MRI and PET sources, denoted as $\boldsymbol{X}_0^1$ and $\boldsymbol{X}_1^1$. The second task has 69 samples with features from MRI

and GENE sources, denoted as $X_0^2$ and $X_2^2$. The MRI source includes 267 features, PET includes 113, and GENE includes 300. For the response variable, we use the Mini-Mental State Examination (MMSE), which measures cognitive impairment and serves as a diagnostic indicator of Alzheimer's disease (Tombaugh & McIntyre, 1992). We provide a detailed description of the ADNI database in Appendix A.5.1

Although both tasks share the MRI source, significant heterogeneity may still exist between the two datasets. To quantitatively assess this heterogeneity, we calculate the Maximum Mean Discrepancy (MMD) distance between $X_0^1$ and $X_0^2$. Additionally, a permutation test is conducted to determine whether the differences between these sample sets are statistically significant. The test yields a $p$-value of $1 \times 10^{-6}$, indicating significant differences between $X_0^1$ and $X_0^2$ and therefore a neces-

| Method | Task 1 | Task 2 |
|--------|--------|--------|
| STL | 2.74(0.87) | 4.57(1.15) |
| HTL | 2.86(0.75) | 4.34(1.47) |
| **Ours** | **2.66**(0.59) | **3.59**(0.98) |

Table 1: Prediction accuracy on testing data, measured by RMSE.

sity of incorporating heterogeneity among homogeneous source in MTL. Furthermore, the small sample sizes in both tasks impose challenges for prediction, where MTL can potentially enhance performance. For both datasets, we use 60% of the samples for training, 20% for model selection and early stopping, and calculate RMSE on the remaining 20% for testing. The experiment is repeated 30 times, and the results are summarized in Table 1. Our MTL-HMB yields lower prediction errors in both tasks, particularly in Task 2, where it improves performance by at least 17.28% compared to the other two methods, despite the small sample sizes. HTL performs worse than STL due to ignoring the significant heterogeneity between $X_0^1$ and $X_0^2$.

Figure 7 presents the t-SNE visualization of the latent representations obtained from a single training session, where the proposed MTL-HMB method effectively captures both shared and task-specific representations. Notably, the shared representations of the two tasks form a single cluster, while the task-specific representations of the two tasks exhibit significant differences in their distributions. This indicates that the datasets for the two tasks share certain commonalities while also displaying clear heterogeneity, which requires careful consideration during integration.

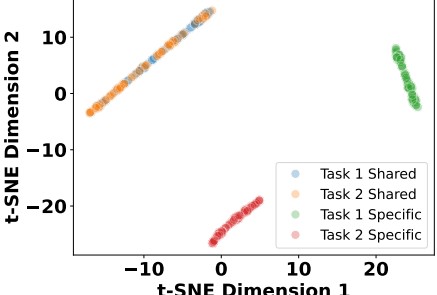

Figure 7: The t-SNE visualization of the learned task-specific and shared representations.

## 5 DISCUSSION

In this paper, we propose a novel two-step strategy for effective MTL in the context of block-wise missing data in conjunction with different types of heterogeneity. The first step addresses distribution heterogeneity using integrated imputation, while the second step integrates learning to overcome distribution and posterior heterogeneity. We conduct extensive numerical experiments to validate the superiority of the proposed method across various levels of heterogeneity. Additionally, in the ADNI real-world dataset, our approach achieves significant improvements in both tasks. In the following, we provide the limitations and outline directions for future work, primarily focusing on transforming the two-step process into a single-step approach. In this unified method, the shared and task-specific hidden representations can be used for both imputing missing data and posterior learning simultaneously, as detailed in Appendix A.6.

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

# A APPENDIX

In Appendix A.1, we further expand on the related works described in Section 2.

In Appendix A.2, we provide the detailed DGP used in Section 4.2.

In Appendix A.3, we conduct ablation experiments to demonstrate the individual roles of the two steps in our proposed method.

In Appendix A.4, we provide the pseudo-code for the proposed MTL-HMB.

In Appendix A.5, we include detailed experimental information, including the real data description and implementation details.

In Appendix A.6, we discuss the limitations of our work and potential future research directions.

In Appendix A.7, we display all qualitative results in Section 4 and compare with additional statistical method.

## A.1 EXPANDED RELATED WORKS

**Multi-group data integration.** Multi-group data integration and MTL share the common goal of learning from multiple datasets or tasks simultaneously. The input features and response of a single task can be viewed as a separate group. There are several existing methods in the statistical literature for multi-group data analysis, which can be broadly classified into three categories. The first category designs specialized regression models (Meinshausen & Bühlmann, 2015; Zhao et al., 2016; Wang et al., 2018; Huang et al., 2023a;b) or factor regression models (Wang et al., 2023a;b) to handle large-scale heterogeneous data and identify group-specific structures. The second category employs specified parameter space constraints, such as fused penalties, to estimate regression coefficients that capture subgroup structures (Tang & Song, 2016; Ma & Huang, 2017; Chen et al., 2021; Li & Sang, 2019; Tang et al., 2021; Lam et al., 2022; Duan & Wang, 2023; Zhang et al., 2024b). The third category involves transfer learning, which borrows information from source data to target data (Li et al., 2022; Tian et al., 2022; Zhang & Zhu, 2022; Tian & Feng, 2023; Cai & Pu, 2024; Cai et al., 2024; He et al., 2024a; Zhang et al., 2024a). The aforementioned multi-group data integration approaches address distribution and posterior heterogeneity but overlook block-wise missing issues. Additionally, most methods rely on structured model assumptions, such as linearity, limiting their capacity to capture complex relationships.

**Heterogeneous feature spaces.** Existing transfer learning methods mainly addressed either distribution shift or posterior shift separately, with fewer studies considering both types of shifts simultaneously. For instance, Moon & Carbonell (2017) investigated scenarios with both heterogeneous feature and label spaces in the context of natural language processing. They proposed a method that learned a common embedding for the features and labels and then established a mapping between them. Similarly, Bica & van der Schaar (2022) focused on a shared label space but assumed that all tasks had a common source, utilizing the same encoder to extract shared representations. However, this assumption was often unrealistic in practice. Even when sources were identical, different tasks could exhibit significant heterogeneity due to variations in subjects, locations, and experimental settings. For example, in our ADNI real data (Section 4.3), tasks sharing MRI features might still differ due to varying experimental conditions. A key distinction in our method is that we treat this problem as a block-wise missing data issue rather than simply considering each task to have only the observed features. This perspective aligns more closely with the reality of medical data, where missing problem is common, and these missing features can also influence the response. Additionally, we focus on MTL, which is designed for numerous small-sample and challenging tasks. In contrast, transfer learning often assumed the existence of a large-scale dataset to support a smaller-sample task. For example, in the experiments conducted by Bica & van der Schaar (2022), the source domain's sample size was typically more than ten times that of the target domain. However, in real-world scenarios, it is more common for all tasks to have relatively small and limited sample sizes. Our method aims to provide a more comprehensive and robust learning framework by integrating heterogeneous information across these small-sample tasks.

**Block-wise Statistical Methods.** Numerous statistical methods for block-wise missing data have been proposed, and we provide a more detailed discussion here. Yu et al. (2020); Wang et al.

(2024a) learn linear predictors through covariance matrix and cross-covariance vector, which can be estimated with block-missing data without imputation. Xue & Qu (2021); Xue et al. (2021) propose a multiple block-wise imputation (MBI) approach to construct estimating equations based on all available information and integrate estimating functions to achieve efficient estimators. Li et al. (2024b) leverages block-wise missing labeled samples and further enhances estimation efficiency by incorporating large unlabeled samples through imputation and projection. Their method is robust to model misspecification on the missing covariates. Song et al. (2024) address a similar problem under the semi-supervised learning setting, employing a double debiased procedure without relying on imputation. Zhou et al. (2023) develop an efficient block-wise overlapping noisy matrix integration algorithm to obtain multi-source embeddings. These methods have demonstrated strong performance in various real-world applications. For instance, Zhou et al. (2023); Li et al. (2024b) validated their methods on electronic health record (EHR) data, demonstrating their effectiveness in real-world applications. However, all the aforementioned methods suffer from several limitations. First, they primarily capture linear relationships and struggle to effectively learn nonlinear patterns. Many real-world datasets, such as multi-modal single-cell data (Tu et al., 2022; Cohen Kalafut et al., 2023) and imaging data (Jin et al., 2017; Bernal et al., 2019), exhibit complexities that further limit the applicability of these methods. This limitation underscores the motivation for adopting an encoder-decoder framework in our work. Second, these methods consider the homogeneous model setup for different tasks, for instance, assuming the same regression coefficients are applied to all tasks. However, data heterogeneity across tasks or sources are ubiquitous in real applications, either marginal distribution of sources or conditional distribution among sources can be distinct, which complicates the modeling procedure. This is another motivation of our project, to effectively handle multiple types of heterogeneity simultaneously.

## A.2 DATA GENERATION PROCESS IN SECTION 4.2

We consider MTL for multiple tasks. The DGP is similar to that in Section 4.1 but is extended to accommodate more tasks. For three-task learning, the features for the $t$-th task are denoted as $\boldsymbol{x}^t = [\boldsymbol{x}_0^t | \boldsymbol{x}_1^t | \boldsymbol{x}_2^t | \boldsymbol{x}_3^t]$ and follow a Gaussian distribution with mean $\boldsymbol{0}$ and an exchangeable covariance matrix. The variance is fixed at 1, and the covariance structure is determined by $(\rho_t)^{0.01|i-j|}$. We randomly generate $n_t$ samples, with only $\boldsymbol{x}_0^t$ and $\boldsymbol{x}_t^t$ being observed. The response $y^t$ is given by:

$$y^t = \alpha \sum_{d=1}^{p} v_{c,d}(x_d^t)^2/p + (1-\alpha)\sum_{d=1}^{p} v_{t,d}x_d^t/p + \varepsilon_r, \quad \forall r \in [3].$$

where $p = \sum_{s=0}^{3} p_s$. For three-task learning, we choose the following parameters: $n_1 = n_2 = n_3 = 300$, $p_0 = 125$, $p_1 = p_2 = p_3 = 25$, $\rho_1 = 0.95$, $\rho_2 = \rho_3 = 0.9$, $\alpha = 0.3$, $v_c, v_t \sim N(-10, 10^2)$, and $\varepsilon_t \sim N(0, 0.01)$ for $t \in [3]$.

For four-task learning, the features for the $t$-th task are denoted as $\boldsymbol{x}^t = [\boldsymbol{x}_0^t | \boldsymbol{x}_1^t | \boldsymbol{x}_2^t | \boldsymbol{x}_3^t | \boldsymbol{x}_4^t]$ and follow a Gaussian distribution with mean $\boldsymbol{0}$ and an exchangeable covariance matrix. The variance is fixed at 1, and the covariance structure is determined by $(\rho_t)^{0.01|i-j|}$. We randomly generate $n_t$ samples, with only $\boldsymbol{x}_0^t$ and $\boldsymbol{x}_t^t$ being observed. The response $y^t$ is given by:

$$y^t = \alpha \sum_{d=1}^{p} v_{c,d}(x_d^t)^2/p + (1-\alpha)\sum_{d=1}^{p} v_{t,d}x_d^t/p + \varepsilon_r, \quad \forall r \in [4].$$

where $p = \sum_{s=0}^{4} p_s$. For four-task learning, we choose the following parameters: $n_1 = n_2 = n_3 = n_4 = 300$, $p_0 = 125$, $p_1 = p_2 = p_3 = p_4 = 25$, $\rho_1 = 0.95$, $\rho_2 = \rho_3 = \rho_4 = 0.9$, $\alpha = 0.3$, $v_c, v_t \sim N(-10, 10^2)$, and $\varepsilon_t \sim N(0, 0.01)$ for $t \in [4]$.

For evaluation, we focus on the average RMSE across all tasks in the testing data, defined as follows:

$$\text{RMSE} = \frac{1}{T}\sum_{t=1}^{T}\sqrt{\frac{1}{n_{t,\text{test}}}\sum_{i=1}^{n_{t,\text{test}}}(\widehat{y}_i^t - y_i^t)^2}.$$

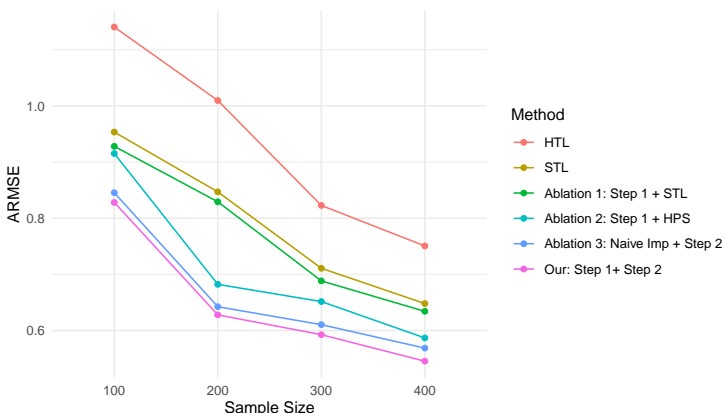

Figure 8: The average RMSEs of all methods across different $n_2$ sample sizes. HPS refers to hard parameter sharing, and Imp refers to imputation.

### A.3 ABLATION EXPERIMENTS

We propose an MTL framework that involves two steps: Step 1 for HBI (see Section 3.1) and Step 2 for heterogeneous MTL (see Section 3.2). To assess the independent effect of each step, we design ablation experiments. In addition to comparing with STL and HTL, we consider three new ablation experiments. The first is Step 1 + STL, which applies HBI followed by STL to evaluate the effect of Step 2 and is denoted as Ablation 1. The second approach is Step 1 combined with a common MTL framework (Ablation 2). Specifically, we adopt the hard parameter sharing (HPS) framework, which shares the main layers across tasks while differentiating in the final layer, and is widely used in MTL (Liu et al., 2019; Bai et al., 2022). However, hard parameter sharing struggles to address distribution heterogeneity due to the shared structure in the first $L-1$ layers. The third is naive imputation + Step 2, where we ignore distribution heterogeneity to analyze the impact of disregarding heterogeneity in imputation, denoted as Ablation 3.

The data generation process is consistent with Section 4.1, but we adopt a more challenging setting. Specifically, we set $p_1 = 100$, $p_2 = 25$, $p_3 = 25$, $\rho_1 = 0.8$, $\rho_2 = 0.6$, $\alpha = 0.3$, and $\sigma_1 = \sigma_2 = 0.1$. We analyze the impact of sample sizes $n_1$ and $n_2$ on three methods by fixing $n_1 = 300$ and varying $n_2$ as $n_2 = k \times 100$ for $k = 1, \ldots, 4$. The experiments are repeated 30 times, and the mean RMSE per task is computed, with the results summarized in Figure 8. It is important to note that, due to the presence of distribution heterogeneity in this setting, HTL performs the worst.

We analyze the ablation results from different perspectives. First, it is evident that both Ablation 3 and our proposed MTL-HMB outperform STL, Ablation 1, and Ablation 2, indicating that Step 2 plays a crucial role in enhancing prediction performance. Second, by comparing Ablation 1 with STL, we observe that Ablation 1 consistently achieves lower loss across different sample sizes, demonstrating that Step 1 improves predictions for a single dataset. Third, when comparing Ablation 3 with our proposed method, Ablation 3 shows higher loss, suggesting that ignoring distribution heterogeneity in imputation negatively impacts performance. Fourth, we compare Ablation 1, Ablation 2, and our proposed MTL-HMB method, all of which incorporate Step 1. The prediction results demonstrate that our method outperforms both Ablation 2 and Ablation 1. This indicates that our MTL framework in Step 2 is more effective than hard parameter sharing, as it accounts for distribution heterogeneity, while hard parameter sharing performs better than STL. Fifth, even when comparing Ablation 2 with Ablation 3—which uses a less effective imputation method—the latter still achieves better predictive performance. This further highlights the advantages of Step 2 over traditional MTL approaches. Overall, the ablation experiments demonstrate that when both distribution and posterior heterogeneity are present, both steps of our proposed framework are crucial.

### A.4 PSEUDO-CODE FOR OUR PROPOSED MTL-HMB

Algorithm 1 provides the pseudo-code for training our proposed MTL method. For simplicity, we set the mini-batch size to be the same across all $T$ datasets: $B^t = B$ for $t \in [T]$. For HBI, we divide the

data into training and testing sets and train the parameters on the training set. Early stopping is applied to $\mathcal{L}_{\text{pre}}$ on the $t$-th dataset's testing data to check for convergence and perform model selection. For heterogeneous MTL, the data is split into training, validation, and testing sets. Parameters are trained on the training set, and the best hyperparameter combination is selected using the validation set. Early stopping is applied to $\mathcal{L}_{\text{integ}}$ on the validation set to check for convergence, and the final prediction metrics are calculated on the test set. In practice, we choose the regularization parameters $\gamma$, $\delta$, and $\kappa$ from the set $[0.01, 0.1, 1]$ for $\mathcal{R}_{\text{orth}}$, $\mathcal{R}_{\text{imp}}$, and $\mathcal{R}_{\text{dr}}$. In our experiments, we found that the selection of $\gamma$, $\delta$, and $\kappa$ is robust, having minimal impact on the final prediction performance.

## A.5 Experimental details

### A.5.1 Dataset description

In this subsection, we provide a detailed description of the ADNI database used in Section 4.3. The ADNI study (Mueller et al., 2005a) aims to identify biomarkers that track the progression of Alzheimer's disease (AD). The MMSE score, which measures cognitive impairment, is treated as the response variable, and we aim to select biomarkers from three complementary data sources: MRI, PET, and gene expression. Given the sparsity assumption, we use region of interest (ROI) level data rather than raw imaging data, as the latter might not be suitable for our method. MRI variables include volumes, cortical thickness, and surface areas, while PET features represent standard uptake value ratios (SUVR) of different ROIs. Gene expression variables are derived from blood samples and represent expression levels at different gene probes. To reduce the number of gene expression variables, we apply sure independence screening (SIS), narrowing it down to 300 variables. This results in a total of 680 features, including 267 MRI features and 113 PET features. The data is sourced from ADNI-2 at month 48, where block-wise missingness occurs due to factors such as low-quality images or patient dropout. Using visit codes, we align MMSE with the imaging data to ensure they are measured within the same month. Ultimately, we obtained two datasets: dataset 1 contains only MRI and PET sources, while dataset 2 includes MRI and gene expression sources. Both datasets have relatively small sample sizes, underscoring the importance of effectively using incomplete observations in the analysis.

### A.5.2 Implementation details and hyperparameter tuning

In Section 4, we compare our proposed method (MTL-HMB) with Single Task Learning (STL) and Heterogeneous Transfer Learning (HTL). Here, we provide the implementation details of these three methods. For STL, we use standard deep neural networks to train each dataset individually. In contrast, HTL assumes no heterogeneity in the anchoring source and extracts task-shared representations from it, while task-specific representations are derived from task-specific sources.

**STL.** For STL, we use standard deep neural networks to train each dataset individually. Each dataset is split into 60% for training, 20% for validation, and 20% for testing. On the training set, we perform hyperparameter tuning, including network width from $\{32, 64, 128\}$, depth from $\{2, 3, 4, 5\}$, and batch size from $\{8, 16, 32\}$ (with 8 included due to the smaller sample size in the ADNI database). Additionally, we set the learning rate to 0.001 and the early-stopping patience to 30. To stabilize the optimization during iterations, we use the exponential scheduler (Patterson & Gibson, 2017), which decays the learning rate by a constant per epoch. In all numerical tasks, we set the decay constant to 0.95, applied every 200 iterations. We tune the hyperparameters and select the best model on the validation set. Finally, the tuned hyperparameters are used to compute the prediction loss on the testing set.

**HTL.** For HTL, we adapt the network architecture from Bica & van der Schaar (2022) and modify it for our setting. Following their approach, the framework for handling heterogeneous feature spaces consists of a common encoder for shared source and task-specific encoders for task-specific sources, implemented using deep neural networks. The network widths are selected from $\{32, 64, 128\}$ and depths from $\{2, 3, 4\}$. The remaining components are incorporated into an MTL network architecture, similar to the structure described in Section 3.2, where shared and task-specific pathways have depths chosen from $\{2, 3, 4\}$. The output dimensions of the first $L - 1$ layers are selected from $\{32, 64, 128\}$, with the final layer predicting the corresponding response. The batch size is chosen from $\{8, 16, 32\}$, and the learning rate is set to 0.001. To remain consistent with Bica & van der Schaar (2022), we train the prediction loss on the training set, along with regularization

terms $\mathcal{R}_{\text{orth}}$ and $\mathcal{R}_{\text{dr}}$. Early stopping and hyperparameter tuning are performed based on the sum of the prediction losses across all datasets on the validation set, with an early-stopping patience of 30. Finally, the tuned hyperparameters are used to compute the prediction loss on the testing set.

**MTL-HMB.** For the proposed MTL-HMB, we describe the method in two steps: Step 1 and Step 2. **Step 1:** In HBI, the common encoder, task-specific encoders, decoder, and predictor use network architectures with widths selected from $\{8, 16, 32\}$ and depths from $\{1, 2, 3\}$. The batch size is chosen from $\{8, 16, 32\}$, and the learning rate is set to $0.001$. Notably, since the features in our simulated data exhibit relatively simple linear relationships, we include smaller network widths and depths in our tuning. **Step 2:** To construct task-shared and task-specific mappings, the network architecture for the shared encoder $\phi_c$ and the task-specific encoders $\phi_p^t$ have widths selected from $\{32, 64, 128\}$ and depths from $\{2, 3, 4\}$. The output dimensions are also chosen from $\{32, 64, 128\}$. For the prediction function, both shared and task-specific pathways have depths chosen from $\{2, 3, 4\}$, with the output dimensions of the first $L-1$ layers selected from $\{32, 64, 128\}$. The final layer predicts the corresponding response. The batch size is chosen from $\{8, 16, 32\}$, and the learning rate is set to $0.001$. Early stopping and hyperparameter tuning are conducted based on the sum of the prediction losses across all datasets on the validation set, using an early-stopping patience of 30. Finally, the tuned hyperparameters are applied to compute the prediction loss on the testing set.

## A.6 DISCUSSION ABOUT LIMITATIONS AND FUTURE WORK

First, our proposed method essentially assumes that there is common information across all tasks that can be fused, which implies a relatively strong shared structure. For example, in Section 3.2, we assume the existence of a common mapping, $\psi_c$, between input features and responses for all $r \in [R]$. However, in reality, when there is strong heterogeneity across multiple datasets, the shared structure is often only partial. For instance, in three datasets, only two may share the common $\psi_c$, while the third task may be too heterogeneous to fuse with the first two. In such cases, an adaptive approach for MTL is needed, one that explores partially shared information among tasks while preserving the uniqueness of the highly heterogeneous task. Currently, some studies have considered adaptive MTL in relatively simple settings, such as linear cases (Duan & Wang, 2023; Tian et al., 2023). However, adaptive MTL in the presence of block-wise, distribution, and posterior heterogeneity remains unexplored, making it a meaningful direction for future research.

Moreover, it is worth noting that in both Section 3.1 (HBI) and Section 3.2 (MTL), the hidden representations in each dataset are learned in two steps: the first step for imputation and the second step for learning the response. This process introduces some computational redundancy. A possible improvement would be to combine these two steps into one, unifying multiple tasks to learn the hidden representations for each task, which can then be used for both imputation and response learning. However, this approach poses computational challenges, such as how to balance different loss functions to achieve both accurate imputation and prediction. Thus, this remains a future research direction worth exploring.

## A.7 QUALITATIVE RESULTS

As suggested by the reviewers, we display all qualitative results from Section 4 in this subsection. Specifically, we report the means and standard deviations of the average RMSEs under different experimental settings. Additionally, we apply the statistical block-wise imputation method (MBI) proposed by Xue & Qu (2021); Xue et al. (2021) to various simulation settings and the ADNI real dataset. In particular, MBI does not account for distribution or posterior heterogeneity. It assumes that the relationships among all sources across different tasks are consistent, as well as the relationships between the sources and the response. The method first imputes all missing blocks and then constructs estimating equations based on the available information. These estimating equations are subsequently integrated to achieve efficient estimators. We used the R package `BlockMissingData` to conduct the experiments, with the tuning parameters set to their default values. The RMSE was computed on a $20\%$ testing set.

Tables 2 to 7 present the prediction results under settings A to F in Section 4.1, while Table 8 corresponds to the results in Section 4.2. We have included the prediction results of the MBI method for comparison. The performance of STL, HTL, and the proposed MTL-HMB methods is extensively discussed in Section 4, where it is evident that MBI significantly underperforms compared to

these three methods. For example, as shown in Table 2, the prediction error of MBI is several times higher than that of the proposed MTL-HMB method. This poor performance can be attributed to several factors. First, MBI cannot handle nonlinear relationships and is limited to modeling linear relationships between sources and the response, which severely restricts its learning capacity. These findings underscore the substantial benefits of leveraging the encoder-decoder framework. Second, MBI is unable to address distribution or posterior heterogeneity.

To ensure a fair comparison, we reconsidered a linear data-generating process (DGP). Specifically, we modified the nonlinear DGP described in Section 4.1 to a simpler linear DGP as follows:

$$y^1 = \alpha \sum_{d=1}^{p} v_{c,d} x_d^1 / p + (1 - \alpha) \sum_{d=1}^{p} v_{1,d} x_d^1 / p + \varepsilon_1,$$

$$y^2 = \alpha \sum_{d=1}^{p} v_{c,d} x_d^2 / p + (1 - \alpha) \sum_{d=1}^{p} v_{2,d} x_d^2 / p + \varepsilon_2,$$

Where other parameters and settings remain unchanged, we evaluated the prediction performance of the four methods under this linear DGP. The results, presented in Table 9, show that MTL-HMB still achieves the best performance, followed by STL. HTL is constrained by distribution heterogeneity, while MBI, despite being designed for linear cases, suffers significant errors starting from the imputation step due to its assumption of no distribution or posterior heterogeneity. Consequently, its final predictions are notably poor.

Additionally, we evaluated MBI on the ADNI real dataset. The prediction results for Task 1 and Task 2 were $9.847, (3.516)$ and $10.272, (3.448)$, respectively. These findings further demonstrate the significant improvements brought by the encoder-decoder framework in real-world applications.

Table 2: Average RMSEs under **Setting A**.

| $\rho_1 = \rho_2$ | STL | HTL | MTL-HMB | MBI |
|---|---|---|---|---|
| 0.5 | 0.650(0.116) | 0.593(0.130) | 0.593(0.188) | 5.155(0.936) |
| 0.6 | 0.604(0.150) | 0.529(0.114) | 0.474(0.098) | 5.089(0.936) |
| 0.7 | 0.535(0.170) | 0.434(0.077) | 0.421(0.107) | 4.921(0.842) |
| 0.8 | 0.558(0.196) | 0.452(0.098) | 0.382(0.118) | 4.782(0.821) |
| 0.9 | 0.421(0.155) | 0.463(0.138) | 0.345(0.125) | 4.651(0.779) |
| 0.95 | 0.376(0.169) | 0.413(0.182) | 0.270(0.064) | 4.516(0.696) |

Table 3: Average RMSEs under **Setting B**.

| $\rho_1 \neq \rho_2$ | STL | HTL | MTL-HMB | MBI |
|---|---|---|---|---|
| 0.5 | 0.529(0.145) | 0.476(0.128) | 0.410(0.131) | 5.125(0.961) |
| 0.6 | 0.485(0.170) | 0.466(0.140) | 0.378(0.128) | 5.013(0.849) |
| 0.7 | 0.489(0.129) | 0.485(0.116) | 0.375(0.143) | 4.966(0.856) |
| 0.8 | 0.457(0.154) | 0.463(0.109) | 0.333(0.116) | 4.902(0.821) |
| 0.9 | 0.444(0.200) | 0.402(0.127) | 0.314(0.126) | 4.807(0.806) |

Table 4: Average RMSEs under **Setting C**.

| $\alpha$ | STL | HTL | MTL-HMB | MBI |
|---|---|---|---|---|
| 0.1 | 0.218(0.056) | 0.214(0.035) | 0.191(0.050) | 1.825(0.336) |
| 0.2 | 0.366(0.093) | 0.372(0.134) | 0.300(0.085) | 3.356(0.564) |
| 0.3 | 0.558(0.196) | 0.452(0.098) | 0.382(0.118) | 4.921(0.842) |
| 0.4 | 0.585(0.154) | 0.587(0.192) | 0.511(0.207) | 6.551(1.180) |
| 0.5 | 0.810(0.280) | 0.676(0.130) | 0.583(0.192) | 8.152(1.424) |
| 0.6 | 0.970(0.246) | 0.871(0.237) | 0.809(0.266) | 9.711(1.668) |

Table 5: Average RMSEs under **Setting D**.

| $n_1 = n_2$ | STL | HTL | MTL-HMB | MBI |
|---|---|---|---|---|
| 100 | 0.893(0.403) | 0.797(0.294) | 0.598(0.277) | 4.474(1.118) |
| 200 | 0.526(0.141) | 0.532(0.112) | 0.452(0.262) | 4.362(0.803) |
| 300 | 0.489(0.129) | 0.485(0.116) | 0.375(0.143) | 4.966(0.856) |
| 400 | 0.365(0.096) | 0.400(0.105) | 0.327(0.084) | 5.091(0.726) |
| 500 | 0.332(0.066) | 0.367(0.102) | 0.288(0.046) | 5.140(0.627) |
| 600 | 0.303(0.060) | 0.350(0.064) | 0.276(0.055) | 4.867(0.693) |

Table 6: Average RMSEs under **Setting E**.

| $p_1 = p_2$ | STL | HTL | MTL-HMB | MBI |
|---|---|---|---|---|
| 25 | 0.489(0.129) | 0.485(0.116) | 0.375(0.143) | 4.966(0.856) |
| 50 | 0.660(0.141) | 0.658(0.232) | 0.531(0.170) | 4.414(0.874) |
| 75 | 0.712(0.175) | 0.649(0.168) | 0.585(0.161) | 4.745(0.548) |
| 100 | 0.786(0.201) | 0.886(0.208) | 0.692(0.118) | 4.709(0.770) |

Table 7: Average RMSEs under **Setting F**.

| $\sigma_1$ | STL | HTL | MTL-HMB | MBI |
|---|---|---|---|---|
| 0.1 | 0.489(0.129) | 0.485(0.116) | 0.375(0.143) | 4.966(0.856) |
| 0.2 | 0.499(0.156) | 0.438(0.071) | 0.389(0.098) | 4.961(0.877) |
| 0.3 | 0.536(0.119) | 0.522(0.102) | 0.454(0.093) | 4.938(0.858) |
| 0.4 | 0.543(0.120) | 0.569(0.131) | 0.466(0.105) | 4.974(0.872) |
| 0.5 | 0.663(0.162) | 0.647(0.160) | 0.511(0.082) | 5.002(0.855) |

Table 8: Average RMSEs under multiple tasks.

| $T$ | SDL | HTL | Proposed | MBI |
|---|---|---|---|---|
| 2 | 0.429(0.209) | 0.398(0.150) | 0.310(0.111) | 4.776(0.688) |
| 3 | 0.468(0.129) | 0.394(0.109) | 0.344(0.074) | 5.861(0.878) |
| 4 | 0.490(0.116) | 0.451(0.066) | 0.410(0.044) | 6.902(0.976) |

Table 9: Average RMSEs under linear setting.

| STL | HTL | MTL-HMB | MBI |
|---|---|---|---|
| 0.295(0.028) | 0.765(0.167) | 0.274(0.029) | 0.525(0.296) |

---

**Algorithm 1** Pseudo-code for Our Proposed MTL-HMB.

---

1: Input: $T$ datasets denoted by $\{\boldsymbol{x}_i^t, y_i^t\}_{i=1}^{n_t}$, where $\boldsymbol{x}_i^t$ includes two blocks $\boldsymbol{x}_{0,i}^t$ and $\boldsymbol{x}_{t,i}^t$, learning rate $\eta$, mini-batch size for the $t$-th dataset is denoted by $B^t$.

2: **Step 1: HBI**

3: **for** $t = 1, \ldots, T$ **do**                                             ▷ Imputation for task $t$-specific source

4:      Initialize: $\boldsymbol{\theta}^t$ (all parameters in this step)

5:      **while** not converged **do**

6:          Sample mini-batch of $B^t$ demonstrations from the $t$-th dataset $\{\boldsymbol{x}_i^t, y_i^t\}_{i=1}^{n_t}$ and mini-batch combination of $B^{-t} = \sum_{s \neq t} B^t$ demonstrations from the rest $T - 1$ datasets.

7:          **for** $i = 1, \ldots, B^t$ **do**                      ▷ Process batch from the $t$-th dataset.

8:              $\boldsymbol{f}_i^t = E_c(\boldsymbol{x}_{0,i}^t), \boldsymbol{g}_i^t = E_p^t(\boldsymbol{x}_{0,i}^t)$

9:          **end for**

10:         Compute prediction loss $\mathcal{L}_{\text{pre}}^t = \sum_{i=1}^{B^t} l(\boldsymbol{x}_{t,i}^t, G(\boldsymbol{f}_i^t))$

11:         Compute reconstruction loss $\mathcal{L}_{\text{recon}}^t = \sum_{i=1}^{B^t} l(\boldsymbol{x}_{0,i}^t, D(\boldsymbol{f}_i^t, \boldsymbol{g}_i^t))$

12:         **for** $i = 1, \ldots, B^{-t}$ **do**            ▷ Process batch from the rest $T - 1$ datasets.

13:             $\boldsymbol{f}_i^{-t} = E_c(\boldsymbol{x}_{0,i}^{-t}), \boldsymbol{g}_i^{-t} = E_p^{-t}(\boldsymbol{x}_{0,i}^{-t})$

14:         **end for**

15:         Compute reconstruction loss: $\mathcal{L}_{\text{recon}}^{-t} = \sum_{i=1}^{B^{-t}} l(\boldsymbol{x}_{0,i}^{-t}, D(\boldsymbol{f}_i^{-t}, \boldsymbol{g}_i^{-t}))$

16:         Parameter update $\boldsymbol{\theta}^t \leftarrow \boldsymbol{\theta}^t - \eta \nabla_{\boldsymbol{\theta}^t}(\mathcal{L}_{\text{pre}}^t + \mathcal{L}_{\text{recon}}^t + \mathcal{L}_{\text{recon}}^{-t})$

17:      **end while**

18:      **for** $i = 1, \ldots, B^{-t}$ **do**

19:         Imputation for task $t$-specific source: $\widehat{\boldsymbol{x}}_{t,i}^{-t} = \widehat{G}(\widehat{E}_c(\boldsymbol{x}_{0,i}^{-t}))$

20:      **end for**

21: **end for**

22: Obtain samples with reconstructed features $\{(\boldsymbol{x}_{0,i}^t, \ldots, \widehat{\boldsymbol{x}}_{t-1,i}^t, \boldsymbol{x}_{t,i}^t, \widehat{\boldsymbol{x}}_{t+1,i}^t, \ldots, \widehat{\boldsymbol{x}}_{T,i}^t), y_i^t)\}_{i=1}^{n_t}$

23: **Step 2: Heterogeneous MTL**

24: Initialize: $\Theta$ (all parameters in this step)

25: **while** not converged **do**

26:      **for** $t = 1, \ldots, T$ **do**

27:         **for** $i = 1 \ldots B^t$ **do**                    ▷ Process batch from the $r$-th dataset.

28:            $\boldsymbol{h}_i^t = \phi_c(\boldsymbol{x}_{0,i}^t), \boldsymbol{k}_i^t = \phi_p^t([\boldsymbol{x}_{0,i}^t|\cdots|\widehat{\boldsymbol{x}}_{t-1,i}^t|\boldsymbol{x}_{t,i}^t|\widehat{\boldsymbol{x}}_{t+1}^t|\cdots|\widehat{\boldsymbol{x}}_{T,i}^t])$

29:           Set $\boldsymbol{H}^t = [\boldsymbol{h}_i^t \cdots \boldsymbol{h}_{B^t}^t]^\top, \boldsymbol{K}^t = [\boldsymbol{k}_i^t \cdots \boldsymbol{k}_{B^t}^t]^\top$

30:           **for** $l = 1 \ldots L$ **do**

31:             **if** $l == 1$ **then**

32:                $\bar{\boldsymbol{h}}_{l,i}^t = \boldsymbol{h}_i^t, \bar{\boldsymbol{k}}_{l,i}^t = [\boldsymbol{h}_i^t|\boldsymbol{k}_i^t]$

33:             **else**

34:                $\bar{\boldsymbol{h}}_{l,i}^t = \boldsymbol{h}_{l-1,i}^t, \bar{\boldsymbol{k}}_{l,i}^t = [\boldsymbol{h}_{l-1,i}^t|\boldsymbol{k}_{l-1,i}^t]$

35:                $\boldsymbol{h}_{l,i}^t = \text{Shared\_Path}(\bar{\boldsymbol{h}}_{l,i}^t), \boldsymbol{k}_{l,i}^t = \text{Task\_Specific\_Path}^t(\bar{\boldsymbol{k}}_{l,i}^t)$

36:             **end if**

37:           **end for**

38:           $\widehat{y}_i^t = g^t([\boldsymbol{h}_{L,i}^t|\boldsymbol{k}_{L,i}^t])$

39:         **end for**

40:      **end for**

41:      Compute integration loss: $\mathcal{L}_{\text{integ}} = \sum_{t=1}^{T} \sum_{i=1}^{B^t} l(y_i^t, \widehat{y}_i^t)$

42:      Compute orthogonal regularizer for features: $\mathcal{R}_{\text{orth}} = \sum_{t=1}^{T} \|(\boldsymbol{H}^t)^\top \boldsymbol{K}^t\|_F^2$

43:      Compute robust regularizer for imputation: $\mathcal{R}_{\text{imp}} = \sum_{t=1}^{T} \sum_{s \neq 0,t} \|\Theta_{s,p,1}^t\|_F^2$

44:      Compute regularizater for redundancy: $\mathcal{R}_{\text{dr}} = \sum_{t=1}^{T} \sum_{l=1}^{L} \|(\Theta_{c,l}^t)^\top \Theta_{p,l,1:d_{c,l-1}^r}^t\|_F^2$

45:      Parameters update:

46:      $\Theta \leftarrow \Theta - \eta \nabla_{\Theta}(\mathcal{L}_{\text{integ}} + \mathcal{R}_{\text{orth}} + \mathcal{R}_{\text{imp}} + \mathcal{R}_{\text{dr}})$

47: **end while**

48: **Output:** Learnt parameters $\Theta$

---

