# OpenReview forum: "Multi-task Learning for Heterogeneous Multi-source Block-Wise Missing Data"
_ICLR.cc/2025/Conference — Submitted to ICLR 2025_

### Official Review · Reviewer_S8oQ · 2024-10-28

**Soundness:** 3
**Presentation:** 4
**Contribution:** 3
**Rating:** 6
**Confidence:** 3

**Summary:**

This paper proposes a method to perform multi-task learning in a context of heterogeneous multi-source block-wise missing data. The authors propose a block-wise imputation method and then an algorithm designed for heterogeneous multi-task learning. The method is assessed on synthetic data and on the ADNI dataset.

I'm a beginner in multi-task learning and I'm not able to have an opinion on how this article is positioned in this literature. However, I think this paper is well written and well presented, the experiments are complete (apart I find from the comparison with other methods), the proposed methodology is innovative and of interest.

I'm putting 6 as my initial score because I have a few questions.

**Strengths:**

- Main strenght: the paper is easy to follow, the methodology is clearly explained (Figures 2, 3, 4 are really helpful).

**Weaknesses:**

My major concern is on the experimental study (see Questions).

**Questions:**

Main questions:
1. The whole pipeline can be decomposed into two steps: one imputation step and one prediction step. In the missing-data literature (out of the scope of MTL context), this is recommended to use a two-step procedure when the learning task is prediction but it has been recently shown that naive imputation is adequate. I wonder if here the authors can think about a "naive" imputation in the MTL context.
See for example: Le Morvan, Marine, et al. "What’sa good imputation to predict with missing values?." Advances in Neural Information Processing Systems 34 (2021): 11530-11540.

Experimental study:
1. The authors only compare their methodology to Single Task Learning (STL) and Transfer Learning for Heterogeneous Data (HTL).
- Can they describe more in details these existing methods ?
- For STL, how does it work ? Each task is handled independently and the results are aggregated after that ?
- I am not familiar with the MTL literature, but the authors cite some existing works in Section 2. I understand that there is no existing work which handle both the heterogeneity and the missing data problem, but the authors can for example assess the performance of their block-wise missing imputation method + an existing MTL algorithm or an existing block-wise imputation method + their MTL architecture.
2. How many sources and tasks can the method handle?
3. For me, the validation set should not have block-wise missing data. If true, I think the validation set size is too big (20% for model selection and early stopping + 20% for test set size).
4. One thing that I wonder is how is the test set: does it have any block-wise missing data also or all the sources are observed ?


Some minor comments:
- The notation $\mathcal{L}_{\mathrm{recon}}$ is not introduced in the main text
- In Figure 3, the final arrows are not clear. Why is there this orange arrow ? Why a white case for the third line (imputation case ?) ? And finally, the location of "G" and "D" is unclear for me (even though this is clear in the main text).

---

> ### Author Response · Authors · 2024-11-16
> **Rebuttal 1 by Authors**
>
> Thank you for your constructive feedback. We have considered your comments, revised the paper, and provided a point-by-point response below. Should you have further questions or insights, please let us know.
>
> 1. **if here the authors can think about a "naive" imputation in the MTL context.**
>
>    Thank you for your comment and referring us to that Neurips paper. Considering a naive imputation in MTL with weak heterogeneity is indeed a meaningful direction, as it can significantly reduce computational complexity. However, the problem we address involves MTL with distribution and posterior heterogeneity. In particular, distribution heterogeneity across tasks implies that the correlations between sources within each task may vary. Using a naive imputation method could potentially ignore these heterogeneity, leading to imputation errors.
>
>    In Appendix A.3 (ABALATION EXPERIMENTS), we explored this setting in Ablation 3, where we applied a naive imputation combined with Step 2. The results showed that the final predictions were inferior to those of our proposed method. This underscores the necessity of accounting for heterogeneity during imputation. It is worth noting that Ablation 3 still performed reasonably well, suggesting that naive imputation could be a promising direction for further exploration.
>
> 2. The authors only compare their methodology to Single Task Learning (STL) and Transfer Learning for Heterogeneous Data (HTL).
>
>    Thank you for your comment. Our work focuses on addressing block-wise missing data as well as distribution and posterior heterogeneity in complex settings. Relevant studies in this area are very limited, which is why we primarily compare our method with STL and HTL. To provide a more comprehensive comparison, we have included the performance of the MBI  [1] method across all nonlinear settings in Appendix A.7 (QUALITATIVE RESULTS). We used the R package "BlockMissingData" to conduct the experiments, with tuning parameters set to their default values. The RMSE was computed on a 20% testing set. As expected, the prediction error of MBI is several times higher than that of the proposed MTL-HMB method. This poor performance can be attributed to several factors. First, MBI cannot handle nonlinear relationships and is limited to modeling linear interactions between sources and the response, which significantly restricts its learning capacity. These findings underscore the substantial benefits of leveraging the encoder-decoder framework. Second, MBI is unable to address distribution or posterior heterogeneity. Detailed results can be found in Appendix A.7 (QUALITATIVE RESULTS). For reference, we present the prediction losses under Setting A.
>
>    **Table: Average RMSEs under Setting A.**
>
>    | $\rho_1 = \rho_2 $ | STL           | HTL           | MTL-HMB       | MBI           |
>    | ------------------ | ------------- | ------------- | ------------- | ------------- |
>    | 0.5                | 0.650 (0.116) | 0.593 (0.130) | 0.593 (0.188) | 5.155 (0.936) |
>    | 0.6                | 0.604 (0.150) | 0.529 (0.114) | 0.474 (0.098) | 5.089 (0.936) |
>    | 0.7                | 0.535 (0.170) | 0.434 (0.077) | 0.421 (0.107) | 4.921 (0.842) |
>    | 0.8                | 0.558 (0.196) | 0.452 (0.098) | 0.382 (0.118) | 4.782 (0.821) |
>    | 0.9                | 0.421 (0.155) | 0.463 (0.138) | 0.345 (0.125) | 4.651 (0.779) |
>    | 0.95               | 0.376 (0.169) | 0.413 (0.182) | 0.270 (0.064) | 4.516 (0.696) |
>
>    For a fair comparison, we reconsidered a linear data-generating process (DGP) and evaluated the prediction performance of four methods under this linear DGP. The results indicate that MTL-HMB still achieves the best performance, followed by STL. HTL is limited by distribution heterogeneity, while MBI, although designed for linear cases, suffers significant errors starting from the imputation step due to its assumption of no distribution or posterior heterogeneity. Consequently, its final predictions are notably poor.
>
>    **Table: Average RMSEs under linear setting.**
>
>    | STL           | HTL           | MTL-HMB       | MBI           |
>    | ------------- | ------------- | ------------- | ------------- |
>    | 0.295 (0.028) | 0.765 (0.167) | 0.274 (0.029) | 0.525 (0.296) |
>
>    Additionally, we evaluated MBI on the ADNI real dataset. The prediction results for Task 1 and Task 2 were $9.847 (3.516)$ and $10.272 (3.448)$, respectively. These findings further demonstrate the significant improvements achieved by the encoder-decoder framework in real-world applications.

---

> > ### Author Response · Authors · 2024-11-16
> > **Rebuttal 2 by Authors**
> >
> > 3. Can they describe more in details these existing methods ?
> >
> >    Thank you for your question. We have provided a more detailed discussion of STL and HTL in the manuscript. For STL, standard deep neural networks are used to train each dataset individually. In contrast, HTL assumes no heterogeneity in the anchoring source and extracts task-shared representations from it, while task-specific representations are derived from task-specific sources. Implementation details and hyperparameter tuning for these three methods are provided in Appendix A.5.
> >
> > 4. For STL, how does it work ? Each task is handled independently and the results are aggregated after that ?
> >
> >    Thank you for your question. For STL, each task is handled independently. We use 60% of the dataset for training, 20% for validation to perform hyperparameter tuning and early stopping, and the remaining 20% for testing, where the RMSE is computed. Finally, the RMSEs across all tasks are aggregated, and their average is calculated for visualization purpose
> >
> > 5. The authors can for example assess the performance of their block-wise missing imputation method + an existing MTL algorithm or an existing block-wise imputation method + their MTL architecture.
> >
> >    Thank you for your comment. We have revised Appendix A.3 (ABALATION EXPERIMENTS), where we consider three different ablation settings and compare all six methods as follows:
> >
> >    - **HTL**
> >    - **STL**
> >    - **Ablation 1:** Step 1 + STL
> >    - **Ablation 2:** Step 1 + hard parameter sharing
> >    - **Ablation 3:** Naive imputation + Step 2
> >    - **Our method:** Step 1 + Step 2
> >
> >    The results are presented in **Figure 8 (Page 19)**. We analyze the ablation results from different perspectives:
> >
> >    1. Both Ablation 3 and our proposed MTL-HMB method outperform STL, Ablation 1, and Ablation 2, indicating that Step 2 plays a crucial role in enhancing the performance of STL.
> >    2. By comparing Ablation 1 with STL, we observe that Ablation 1 consistently achieves lower loss across different sample sizes, demonstrating that Step 1 improves predictions for a single dataset.
> >    3. Comparing Ablation 3 with our proposed method, we find that Ablation 3 shows higher loss, suggesting that ignoring distribution heterogeneity in imputation negatively impacts performance.
> >    4. We compare Ablation 1, Ablation 2, and our proposed MTL-HMB method, all of which incorporate Step 1. The results demonstrate that our method outperforms both Ablation 2 and Ablation 1. This indicates that our MTL framework in Step 2 is more effective than hard parameter sharing, as it accounts for distribution heterogeneity, while hard parameter sharing still performs better than STL.
> >    5. Even when comparing Ablation 2 with Ablation 3—which uses a less effective imputation method—the latter still achieves better predictive performance. This further underscores the advantages of our Step 2 framework over traditional MTL approaches.
> >
> >    Overall, the ablation experiments demonstrate that when both distribution and posterior heterogeneity are present, both steps of our proposed framework are crucial.
> >
> > 6. How many sources and tasks can the method handle?
> >
> >    Thank you for your question. Theoretically, our method is capable of handling a large number of sources and tasks. For example, in Section 4.2, we simultaneously address four tasks and five sources, which is highly challenging. However, from a practical computational perspective, increasing the number of tasks and sources amplifies distribution and posterior heterogeneity while reducing the common information shared among the data. This necessitates a larger network capacity and more meticulous tuning to ensure effective training. Addressing these challenges remains a significant challenge in MTL literature.
> >
> > 7. For me, the validation set should not have block-wise missing data. If true, I think the validation set size is too big (20% for model selection and early stopping + 20% for test set size).
> >
> >    Thank you for your comment. In our method, the validation set includes imputed sources, ensuring that no data is missing. This is consistent with the ADNI real data. Additionally, we have indeed  20% of the data for model selection and early stopping, and another 20% for the test set.
> >
> > 8. One thing that I wonder is how is the test set: does it have any block-wise missing data also or all the sources are observed ?
> >
> >    Thank you for your comment. In the testing data, the sources include imputed values, ensuring that no data is missing. This is consistent with the ADNI real data. In the ADNI real data, some sources are initially missing, but after Step 1 (imputation), a complete dataset is obtained. Consequently, in Step 2 (MTL), the testing data includes the sources with imputed values.
> >
> > 7. The notation Lrecon is not introduced in the main text
> >
> >    Thank you for your comment. We introduced $\mathcal{L}_{\text{recon}}$ in the original manuscript on **Page 5, Lines 233–235**.

---

> > > ### Author Response · Authors · 2024-11-16
> > > **Rebuttal 3 by Authors**
> > >
> > > 8. In Figure 3, the final arrows are not clear. Why is there this orange arrow ? Why a white case for the third line (imputation case ?) ? And finally, the location of "G" and "D" is unclear for me (even though this is clear in the main text).
> > >
> > >    Thank you for your comment. We provide a more detailed explanation here:
> > >
> > >    1. The orange arrow illustrates that the predictor $G(\cdot)$ obtained from the second line can be reused in the third line.
> > >    2. The white case in the third line highlights that $x^{-t}_t$ is completely missing, which is the target we aim to impute.
> > >    3. "G" and "D" primarily represent the decoder and predictor, respectively. Specifically,
> > >       - $G(f^t)=\widehat x^t_t$
> > >       - $D(f^t,g^t)=\widehat x^t_0$
> > >       - $D(f^{-t},g^{-t})=\widehat x^{-t}_0$
> > >
> > > [1] Fei Xue, Rong Ma, and Hongzhe Li. Statistical inference for high-dimensional linear regression
> > > with blockwise missing data. arXiv preprint arXiv:2106.03344, 2021.

---

### Official Review · Reviewer_n7ex · 2024-10-30

**Soundness:** 2
**Presentation:** 2
**Contribution:** 2
**Rating:** 5
**Confidence:** 4

**Summary:**

This paper presents a novel two-step strategy for Multi-Task Learning (MTL) addressing the challenges posed by block-wise missing data and various types of heterogeneity. The proposed method's strength lies in its systematic approach to tackling distribution and posterior heterogeneity through integrated imputation and sequential learning. The numerical experiments demonstrate the method's effectiveness across diverse scenarios, providing compelling evidence of its superiority compared to existing techniques. Additionally, the application to the ADNI real-world dataset highlights its practical relevance.

**Strengths:**

The authors conducted comprehensive numerical experiments, validating the efficacy of their method across various levels of heterogeneity. This adds robustness to their claims and provides confidence in the generalizability of the results.

The two-step approach is clearly defined, and the methods used for imputing missing data and disentangling mappings are appropriately justified.

**Weaknesses:**

1.The authors do not provide a detailed explanation of how the proposed method specifically addresses block-wise datasets in the paper.

2.The authors claim in the paper that they use a shared feature extraction encoder and a task-specific feature extraction encoder. What are the differences between these two, and how are they reflected in the methodology?

3.Why are some formulas numbered while others are not? The authors need to revise and check this.

4.The authors propose several loss functions. What is the relationship between these losses, particularly the reconstruction loss on page 5 and the loss function on page 7?

5.A detailed algorithm flowchart needs to be provided.

**Questions:**

See the above Weaknesses.

---

> ### Author Response · Authors · 2024-11-16
> **Rebuttal 1 by Authors**
>
> We greatly appreciate your comments. We have prepared a revised version and provided detailed responses to all comments below. If there are any additional questions, please let us know.
>
> 1. **The authors do not provide a detailed explanation of how the proposed method specifically addresses block-wise datasets in the paper.**
>
>    Thank you so much for your comment. We provide a more detailed explanation of how we handle the heterogeneous missing problem below.
>
>    Suppose we have data from $T$ tasks, with features collected from $T+1$ sources. For all tasks, we assume a common source, called the anchoring source, is observed. Additionally, each task has its own task-specific source, denoted as $x^t_s$ for the $s$-th source in the $t$-th task. Specifically, $x^t_0$ represents the anchoring source observed in the $t$-th task, and $x^t_t$ denotes the task-specific source for the $t$-th task, while $x^t_{s}, \text{ for } s \neq 0, t$ are missing. For the $t$-th task, we observe $n_t$ samples ${[x^t_{0,i} \mid x^t_{t,i}], y^t_i}_{i=1}^{n_t}$. This block-wise missing pattern is common in real-world applications (see **Page 3, Line 141 onwards and Figure 1**).
>
>    To address this block-wise missing problem with $T$ tasks and $T+1$ sources, we impute the task-specific sources in a parallel fashion. For each task-specific source $s \neq 0$, we utilize the anchoring source across all tasks and $x^s_s$ to impute the unobserved blocks $x^t_s, \text{ for }  t \neq s$. Specifically, for the $t$-th source, only the $t$-th task has observed values for the features $x^t_t$. The imputation process leverages the observed $x^t_0$ and $x^t_t$ along with $x^{-t}_0 = \{ x^r_0 \mid r \neq t \}$ to estimate the missing features in the $t$-th source for the other $T-1$ tasks, where $x^{-t}_t = \{x^r_t \mid r \neq t\}$ are unobserved.
>
>    For example, as shown in **Figure 2 (Page 4)**, we use information from $x^1_0$, $x^1_1$, and $x^{-1}_0$ = {$x^2_0, x^3_0, x^4_0$} to impute the missing blocks $x^{-1}_1$ = {$x^2_1, x^3_1, x^4_1$} for the task 1-specific source. To handle this process effectively, we propose the Heterogeneous Block-wise Imputation (HBI) method, which explicitly addresses distribution heterogeneity during imputation. Following a similar approach, we can impute { $\{x^1_2, x^3_2, x^4_2\}$}, {$\{x^1_3, x^2_3, x^4_3\}$}, and {$\{x^1_4, x^2_4, x^3_4\}$}.
>
> 2. **The authors claim in the paper that they use a shared feature extraction encoder and a task-specific feature extraction encoder. What are the differences between these two, and how are they reflected in the methodology?**
>
>    Thank you for your comment. We address the MTL problem with distribution heterogeneity, where tasks exhibit both homogeneous and heterogeneous information. For example, the shared feature extraction encoder captures information shared across all tasks, such as genes or biomarkers that generally influence multiple diseases. In contrast, the task-specific feature extraction encoder learns task-specific heterogeneity, such as certain genes or expression patterns highly correlated with the prediction of Alzheimer’s or diabetes but less relevant for other diseases.
>
>    To help readers better understand this concept, we have added t-SNE visualization results for the ADNI real data application (see **Figure 7, Page 10**). Figure 7 presents the t-SNE visualization of the latent representations obtained from a single training session. Our proposed MTL-HMB method effectively captures both shared and task-specific representations. Notably, the task-specific latent representations of the two tasks display significant differences in their distributions, learned through the task-specific feature extraction encoder.

---

> > ### Author Response · Authors · 2024-11-16
> > **Rebuttal 2 by Authors**
> >
> > 3. **Why are some formulas numbered while others are not? The authors need to revise and check this.**
> >
> >       Thank you for your question. Some equations in the manuscript, such as Equation 1, are referenced later in the text (e.g., **Page 5, Line 225**) and are therefore numbered. Other equations, which are not referenced, have been left unnumbered for simplicity. We apologize for any inconvenience this may have caused.
> >
> >    4. **The authors propose several loss functions. What is the relationship between these losses, particularly the reconstruction loss on page 5 and the loss function on page 7?**
> >
> >       Thank you for your comment. The proposed MTL-HMB method consists of two steps.
> >
> >       In the first step, **Heterogeneous Block-wise Imputation**, the method optimizes a loss function comprising two components: the prediction loss $\mathcal L_{pre}$, which trains the model to predict $x_{t}$ (the target of interest) and is applied only to the $t$-th task, and the reconstruction loss $\mathcal{L}_{\text{recon}}$, which ensures effective representation extraction. The combination of these two losses facilitates more effective imputation.
> >
> >       In the second step, **Heterogeneous Multi-task Learning**, the objective function $\mathcal{L}_{\text{integ}}$ focuses on predicting the response, while the remaining three terms serve as regularization penalties. These penalties address the orthogonality of representations, imputation error, and reduced redundancy between the shared and task-specific layers. Details on tuning the penalty coefficients to improve training performance are provided in Appendix A.4, where we also include the pseudo-code for the proposed MTL-HMB method.
> >
> >    5. **A detailed algorithm flowchart needs to be provided.**
> >
> >       Thank you for your comment. The complete algorithm flowchart was included in the original manuscript. However, due to page limitations, it has been placed on the **last page**. We apologize for any confusion this may have caused.

---

### Official Review · Reviewer_2cL6 · 2024-11-01

**Soundness:** 3
**Presentation:** 3
**Contribution:** 2
**Rating:** 5
**Confidence:** 4

**Summary:**

Comments：
The manuscript introduces a novel two-step learning strategy for multi-task learning (MTL) that effectively addresses multiple forms of heterogeneity, including block-wise, distributional, and posterior heterogeneity. The proposed approach begins by imputing missing blocks using shared representations from homogeneous sources across different tasks, followed by the disentangling of mappings between input features and responses into shared and task-specific components.

Weaknesses:

1.	Why does Equation 1 minimize only L_pre?

2.	The authors utilize a bar chart to display qualitative results, however, quantitative results would be more appropriate, enabling the reader to make numerical comparisons.

3.	The experiments conducted by the authors were insufficient, resulting in an incomplete evaluation of the model's performance.

**Strengths:**

This paper is well-structured and coherent.

**Weaknesses:**

1.Why does Equation 1 minimize only L_pre?

2.The authors utilize a bar chart to display qualitative results, however, quantitative results would be more appropriate, enabling the reader to make numerical comparisons.

3.The experiments conducted by the authors were insufficient, resulting in an incomplete evaluation of the model's performance.

**Questions:**

1.Why does Equation 1 minimize only L_pre?

2.The authors utilize a bar chart to display qualitative results, however, quantitative results would be more appropriate, enabling the reader to make numerical comparisons.

3.The experiments conducted by the authors were insufficient, resulting in an incomplete evaluation of the model's performance.

---

> ### Author Response · Authors · 2024-11-16
> **Rebuttal 1 by Authors**
>
> Thank you for your valuable feedback. We have reviewed all the comments, revised the manuscript accordingly, and included detailed point-by-point responses below. Please feel free to reach out to us if you have any further questions.
>
> 1. Why does Equation 1 minimize only L_pre?
>
>    We sincerely appreciate your comment. Two loss functions are actually minimized simulataneously. To address the confusion, we have revised Equation 1 to {$\mathcal L_{pre}+\mathcal L_{recon}$}  for improved clarity.
>
> 2. Quantitative results would be more appropriate.
>
>    Thank you for your suggestion. To enhance readability, we have presented the prediction results from Section 4 in tables. Due to the page limit in the main text, these tables have been included in Appendix A.7 (QUALITATIVE RESULTS).
>
> 3. The experiments conducted by the authors were insufficient, resulting in an incomplete evaluation of the model's performance.
>
>    Thank you for your comment. In our revision, we have made substantial additions to the experiments to enhance the comprehensiveness of the evaluation. Below, we provide a detailed explanation point by point:
>
>    1. We have carefully revisited the literature on block-wise statistical methods, with a detailed discussion starting on **Page 17, Line 917**, in the blue-highlighted paragraph. We emphasize that these methods have demonstrated strong performance in various real-world applications. For instance, [1,2] validated their methods on electronic health record (EHR) data, demonstrating their effectiveness in practical scenarios. However, all the aforementioned methods suffer from several limitations. First, they primarily capture linear relationships and struggle to effectively learn nonlinear patterns. Many real-world datasets, such as multi-modal single-cell data [3] and imaging data [4], exhibit complexities that further limit the applicability of these methods. This limitation underscores the motivation for adopting an encoder-decoder framework in our work. Second, these methods assume a homogeneous model setup across tasks, such as applying the same regression coefficients to all tasks. However, data heterogeneity across tasks or sources is ubiquitous in real applications. Both the marginal distributions of sources and the conditional distributions among sources can vary, complicating the modeling process. This is another key motivation for our project: to effectively handle multiple types of heterogeneity simultaneously.

---

> > ### Author Response · Authors · 2024-11-16
> > **Rebuttal 2 by Authors**
> >
> > 2. To provide a more comprehensive comparison, we have included the performance of the MBI  [5] method across all nonlinear settings in Appendix A.7 (QUALITATIVE RESULTS). We used the R package "BlockMissingData" to conduct the experiments, with tuning parameters set to their default values. The RMSE was computed on a 20% testing set. As expected, the prediction error of MBI is several times higher than that of the proposed MTL-HMB method. This poor performance can be attributed to several factors. First, MBI cannot handle nonlinear relationships and is limited to modeling linear interactions between sources and the response, which significantly restricts its learning capacity. These findings underscore the substantial benefits of leveraging the encoder-decoder framework. Second, MBI is unable to address distribution or posterior heterogeneity. Detailed results can be found in Appendix A.7 (QUALITATIVE RESULTS). For reference, we present the prediction losses under Setting A.
> >
> >       **Table: Average RMSEs under Setting A.**
> >
> >       | $\rho_1 = \rho_2 $ | STL           | HTL           | MTL-HMB       | MBI           |
> >       | ------------------ | ------------- | ------------- | ------------- | ------------- |
> >       | 0.5                | 0.650 (0.116) | 0.593 (0.130) | 0.593 (0.188) | 5.155 (0.936) |
> >       | 0.6                | 0.604 (0.150) | 0.529 (0.114) | 0.474 (0.098) | 5.089 (0.936) |
> >       | 0.7                | 0.535 (0.170) | 0.434 (0.077) | 0.421 (0.107) | 4.921 (0.842) |
> >       | 0.8                | 0.558 (0.196) | 0.452 (0.098) | 0.382 (0.118) | 4.782 (0.821) |
> >       | 0.9                | 0.421 (0.155) | 0.463 (0.138) | 0.345 (0.125) | 4.651 (0.779) |
> >       | 0.95               | 0.376 (0.169) | 0.413 (0.182) | 0.270 (0.064) | 4.516 (0.696) |
> >
> >       For a fair comparison, we reconsidered a linear data-generating process (DGP) and evaluated the prediction performance of four methods under this linear DGP. The results indicate that MTL-HMB still achieves the best performance, followed by STL. HTL is limited by distribution heterogeneity, while MBI, although designed for linear cases, suffers significant errors starting from the imputation step due to its assumption of no distribution or posterior heterogeneity. Consequently, its final predictions are notably poor.
> >
> >       **Table: Average RMSEs under linear setting.**
> >
> >       | STL           | HTL           | MTL-HMB       | MBI           |
> >       | ------------- | ------------- | ------------- | ------------- |
> >       | 0.295 (0.028) | 0.765 (0.167) | 0.274 (0.029) | 0.525 (0.296) |
> >
> >       Additionally, we evaluated MBI on the ADNI real dataset. The prediction results for Task 1 and Task 2 were $9.847 (3.516)$ and $10.272 (3.448)$, respectively. These findings further demonstrate the significant improvements achieved by the encoder-decoder framework in real-world applications.
> >
> >    3. We have revised Appendix A.3 (ABALATION EXPERIMENTS), where we consider three different ablation settings and compare all six methods as follows:
> >
> >       - **HTL**
> >       - **STL**
> >       - **Ablation 1:** Step 1 + STL
> >       - **Ablation 2:** Step 1 + hard parameter sharing
> >       - **Ablation 3:** Naive imputation + Step 2
> >       - **Our method:** Step 1 + Step 2
> >
> >       The results are presented in **Figure 8 (Page 19)**. We analyze the ablation results from different perspectives:
> >
> >       1. Both Ablation 3 and our proposed MTL-HMB method outperform STL, Ablation 1, and Ablation 2, indicating that Step 2 plays a crucial role in enhancing the performance of STL.
> >       2. By comparing Ablation 1 with STL, we observe that Ablation 1 consistently achieves lower loss across different sample sizes, demonstrating that Step 1 improves predictions for a single dataset.
> >       3. Comparing Ablation 3 with our proposed method, we find that Ablation 3 shows higher loss, suggesting that ignoring distribution heterogeneity in imputation negatively impacts performance.
> >       4. We compare Ablation 1, Ablation 2, and our proposed MTL-HMB method, all of which incorporate Step 1. The results demonstrate that our method outperforms both Ablation 2 and Ablation 1. This indicates that our MTL framework in Step 2 is more effective than hard parameter sharing, as it accounts for distribution heterogeneity, while hard parameter sharing still performs better than STL.
> >       5. Even when comparing Ablation 2 with Ablation 3—which uses a less effective imputation method—the latter still achieves better predictive performance. This further underscores the advantages of our Step 2 framework over traditional MTL approaches.
> >
> >       Overall, the ablation experiments demonstrate that when both distribution and posterior heterogeneity are present, both steps of our proposed framework are crucial for achieving optimal performance.

---

> > > ### Author Response · Authors · 2024-11-16
> > > **Rebuttal 3 by Authors**
> > >
> > > 4. We have added t-SNE visualization results for the ADNI real data application to better illustrate our method (see **Figure 7, Page 10**). Figure 7 presents the t-SNE visualization of the latent representations obtained from a single training session, where our proposed MTL-HMB method effectively captures both shared and task-specific representations. Notably, the task-specific latent representations of the two tasks display significant differences in their distributions.
> > >
> > >    [1] Doudou Zhou, Tianxi Cai, and Junwei Lu. Multi-source learning via completion of block-wise
> > >    overlapping noisy matrices. Journal of Machine Learning Research, 24(221):1–43, 2023.
> > >
> > >    [2] Yiming Li, Xuehan Yang, Ying Wei, and Molei Liu. Adaptive and efficient learning with blockwise
> > >    missing and semi-supervised data. arXiv preprint arXiv:2405.18722, 2024b.
> > >
> > >    [3] Noah Cohen Kalafut, Xiang Huang, and Daifeng Wang. Joint variational autoencoders for multi-
> > >    modal imputation and embedding. Nature Machine Intelligence, 5(6):631–642, 2023.
> > >
> > >    [4] Jose Bernal, Kaisar Kushibar, Daniel S Asfaw, Sergi Valverde, Arnau Oliver, Robert Mart´ı, and
> > >    Xavier Llad´o. Deep convolutional neural networks for brain image analysis on magnetic reso-
> > >    nance imaging: a review. Artificial intelligence in medicine, 95:64–81, 2019.
> > >
> > >    [5] Fei Xue, Rong Ma, and Hongzhe Li. Statistical inference for high-dimensional linear regression
> > >    with blockwise missing data. arXiv preprint arXiv:2106.03344, 2021.

---

### Official Review · Reviewer_Xk9n · 2024-11-03

**Soundness:** 3
**Presentation:** 2
**Contribution:** 2
**Rating:** 5
**Confidence:** 4

**Summary:**

In this work, the authors provide a two-stage algorithm for multi-source multi-task learning with blockwise missing data. The proposed method is assumption light, and allows for complex missingness structures and heterogeneity across sources. The authors demonstrate the effectiveness of their algorithm via simulations and a well-motivated application to a dataset from the Alzheimer's Disease Neuroimaging Initiative.

**Strengths:**

The paper addresses a well-motivated and pervasive problem in large-scale data analysis, namely the integration of block-wise missing data from distinct sources. The proposed imputation method is a novel application of the encoder-decoder framework that, to my knowledge, is new to the missing data literature. Numerical results indicate that this may be a promising approach for imputation.

**Weaknesses:**

The primary weakness of this work is in the evaluation of the proposed MTL-HMB method. The proposed simulation setting (described in Appendix A.2) is far too small and simple to necessitate the heavy machinery used by the proposed method and the STL and HTL methods also applied to the data.

1. The samples sizes are too small relative to the trained neural networks to draw meaningful conclusions from the simulations. This is most evident in Figure 5d: as n grows, the performance of the single-task learning method substantially improves, nearly mimicking the proposed method in the n = 600 setting. The relatively poor performance of STL in particular across the other settings may just be due to high estimation error in learning the neural network.

2. The data generating mechanism outlined in A.2 is a simple linear model. The authors should compare the STL, HTL, and MTL-HMB methods to analogous tools from the statistical literature, especially a standard least squares estimator, a multi-task learning estimator for hetereogenous tasks (such as the ARMUL framework proposed in Duan and Wang 2023 AoS), and a two-stage estimator for blockwise-missing data under linear models such as that provided by Xue, Ma, and Li 2021.

3. As the provided simulation results do not consider any imputation tools other than the proposed method in this paper, it is impossible to determine whether the MTL-HMB is effective at imputation+multi-task learning, or if imputation alone leads to the slight improvement in performance that we see in the paper. While the Ablation 2 experiment attempts to address this, it is still not clear if the use of the encoder-decoder framework is more effective than a simple linear imputation as used in Xue, Ma, and Li 2021. In general, the paper would benefit greatly from more extensive simulation studies that compare the proposed imputation+prediction method to the many methods already studied in the literature, including:

* Li, Y., Yang, X., Wei, Y., & Liu, M. (2024). Adaptive and Efficient Learning with Blockwise Missing and Semi-Supervised Data. arXiv preprint arXiv:2405.18722.
* Xue, F., Ma, R., & Li, H. (2021). Statistical inference for high-dimensional linear regression with blockwise missing data. arXiv preprint arXiv:2106.03344.
* Zhou, D., Cai, T., & Lu, J. (2023). Multi-source learning via completion of block-wise overlapping noisy matrices. Journal of Machine Learning Research, 24(221), 1-43.
* Song, S., Lin, Y., & Zhou, Y. (2024). Semi-supervised Inference for Block-wise Missing Data without Imputation. Journal of Machine Learning Research, 25(99), 1-36.

As it stands, I am unable to evaluate whether the proposed method is meaningful improvement over existing works in this field.

**Questions:**

How does the proposed method perform in larger-scale simulations, or under different (i.e. nonlinear) data-generating models?

---

> ### Author Response · Authors · 2024-11-16
> **Rebuttal 1 by Authors**
>
> Thank you for your thoughtful feedback on our paper. We have carefully reviewed all the comments, provided a revised version, and included point-by-point responses below. If you have any additional questions or further feedback, please don’t hesitate to reach out to us.
>
> 1. **The samples sizes are too small relative to the trained neural networks to draw meaningful conclusions from the simulations.**
>
>    Thank you for your comment. First, we chose small sample sizes in the simulations to better reflect real-world applications. In practical scenarios, such as medical or genomic studies, data collection is often expensive, and sample sizes are typically limited. This constraint motivates the integration of different tasks. For example, in the ADNI real dataset used in our study, two datasets contain only 72 and 69 samples, respectively, which highlights this limitation.
>
>    Second, from a theoretical perspective, the difference between integration and STL diminishes as the sample size increases. Referring to the theoretical results in [1], the worst-case coefficient error for integrating $T$ tasks with sample size $n$ per task is given by $\left( \sqrt{\frac{pr}{nT}} + \sqrt{\frac{r}{n}} + h + \sqrt{\epsilon r} \right) \wedge \sqrt{\frac{p}{n}}$, whereas the rate for STL is $\sqrt{\frac{p}{n}}$. This indicates that as $n$ grows larger, STL increasingly approximates the performance of MTL, particularly when the measures of heterogeneity, $h$ and $\epsilon$, are relatively large. To validate this phaenomenon, we compare STL with our MTL-HMB as the sample size increases to 1000, we found their performance (average RMSE) are similar to each other. Therefore, it is less advantageous to adopt multi-task learning when sample sizes for each task is large.
>
>    | STL           |      | MTL-HMB       |      |
>    | ------------- | ---- | ------------- | ---- |
>    | 0.257 (0.017) |      | 0.256 (0.015) |      |
>
> 2. **The data generating mechanism outlined in A.2 is a simple linear model.**
>
>    We sincerely apologize for any confusion caused. The DGP in our manuscript is indeed nonlinear, as it involves element-wise quadratic terms in generating $y$. To clarify this, we have highlighted the squared terms in blue in the revision. Please refer to **Page 7, Lines 365–370, and Page 18, Lines 948 and 960**.
>
> 3. **In general, the paper would benefit greatly from more extensive simulation studies that compare the proposed imputation+prediction method to the many methods already studied.**
>
>    We appreciate your constructive suggestions. In our revision, we have conducted substantially more simulation experiments to enhance the comprehensiveness of the evaluation. Below, we provide a detailed explanation point by point:
>
>    1. We have carefully revisited the literature on block-wise statistical methods, with a detailed discussion starting on **Page 17, Line 917**, in the blue-highlighted paragraph. We emphasize that these methods have demonstrated strong performance in various real-world applications. For instance, [2,3] validated their methods on electronic health record (EHR) data, demonstrating their effectiveness in practical scenarios. However, all the aforementioned methods suffer from several limitations. First, they primarily capture linear relationships and struggle to effectively learn nonlinear patterns. Many real-world datasets, such as multi-modal single-cell data [4] and imaging data [5], exhibit complexities that further limit the applicability of these methods. This limitation underscores the motivation for adopting an encoder-decoder framework in our work. Second, these methods assume a homogeneous model setup across tasks, such as applying the same regression coefficients to all tasks. However, data heterogeneity across tasks or sources is ubiquitous in real applications. Both the marginal distributions of sources and the conditional distributions among sources can vary, complicating the modeling process. This is another key motivation for our project: to effectively handle multiple types of heterogeneity simultaneously.

---

> > ### Author Response · Authors · 2024-11-16
> > **Rebuttal 2 by Authors**
> >
> > 2. To provide a more comprehensive comparison, we have included the performance of the MBI  [6] method across all nonlinear settings in Appendix A.7 (QUALITATIVE RESULTS). We used the R package "BlockMissingData" to conduct the experiments, with tuning parameters set to their default values. The RMSE was computed on a 20% testing set. As expected, the prediction error of MBI is several times higher than that of the proposed MTL-HMB method. This poor performance can be attributed to several factors. First, MBI cannot handle nonlinear relationship and is limited to modeling linear interactions between sources and the response, which significantly restricts its learning capacity. These findings underscore the substantial benefits of leveraging the encoder-decoder framework. Second, MBI assumes a homogeneous linear regression model across tasks, which is unable to address posterior heterogeneity presented in our simulation setting. Third, due to the distribution heterogeneity across tasks, it is also error-prone to adopt a homogeneous missing imputation approach. Detailed results can be found in Appendix A.7 (QUALITATIVE RESULTS). For reference, we present the prediction losses under Setting A.
> >
> >       **Table: Average RMSEs under Setting A.**
> >
> >       | $\rho_1 = \rho_2 $ | STL           | HTL           | MTL-HMB       | MBI           |
> >       | ------------------ | ------------- | ------------- | ------------- | ------------- |
> >       | 0.5                | 0.650 (0.116) | 0.593 (0.130) | 0.593 (0.188) | 5.155 (0.936) |
> >       | 0.6                | 0.604 (0.150) | 0.529 (0.114) | 0.474 (0.098) | 5.089 (0.936) |
> >       | 0.7                | 0.535 (0.170) | 0.434 (0.077) | 0.421 (0.107) | 4.921 (0.842) |
> >       | 0.8                | 0.558 (0.196) | 0.452 (0.098) | 0.382 (0.118) | 4.782 (0.821) |
> >       | 0.9                | 0.421 (0.155) | 0.463 (0.138) | 0.345 (0.125) | 4.651 (0.779) |
> >       | 0.95               | 0.376 (0.169) | 0.413 (0.182) | 0.270 (0.064) | 4.516 (0.696) |
> >
> >       For a fair comparison, we consider a new linear data-generating process (DGP) and evaluated the prediction performance of four methods under this linear DGP. The results indicate that MTL-HMB still achieves the best performance, followed by STL. HTL is limited by distribution heterogeneity, while MBI, although designed for linear cases, suffers significant errors starting from the imputation step due to its assumption of no distribution or posterior heterogeneity. Consequently, its final predictions are notably poor.
> >
> >       **Table: Average RMSEs under linear setting.**
> >
> >       | STL           | HTL           | MTL-HMB       | MBI           |
> >       | ------------- | ------------- | ------------- | ------------- |
> >       | 0.295 (0.028) | 0.765 (0.167) | 0.274 (0.029) | 0.525 (0.296) |
> >
> >       Additionally, we evaluated MBI on the ADNI real dataset. The prediction results for Task 1 and Task 2 were $9.847 (3.516)$ and $10.272 (3.448)$, respectively. These findings further demonstrate the significant improvements achieved by our proposed method in real-world applications.
> >
> >    3. We have revised Appendix A.3 (ABALATION EXPERIMENTS), where we consider three different ablation settings and compare all six methods as follows:
> >
> >       - **HTL**
> >       - **STL**
> >       - **Ablation 1:** Step 1 + STL
> >       - **Ablation 2:** Step 1 + hard parameter sharing
> >       - **Ablation 3:** Naive imputation + Step 2
> >       - **Our method:** Step 1 + Step 2
> >
> >       The results are presented in **Figure 8 (Page 19)**. We analyze the ablation results from different perspectives:
> >
> >       1. Both Ablation 3 and our proposed MTL-HMB method outperform STL, Ablation 1, and Ablation 2, indicating that Step 2 plays a crucial role in enhancing the performance of STL.
> >       2. By comparing Ablation 1 with STL, we observe that Ablation 1 consistently achieves lower loss across different sample sizes, demonstrating that Step 1 improves predictions for a single dataset.
> >       3. Comparing Ablation 3 with our proposed method, we find that Ablation 3 shows higher loss, suggesting that ignoring distribution heterogeneity in imputation negatively impacts performance.
> >       4. We compare Ablation 1, Ablation 2, and our proposed MTL-HMB method, all of which incorporate Step 1. The results demonstrate that our method outperforms both Ablation 2 and Ablation 1. This indicates that our MTL framework in Step 2 is more effective than hard parameter sharing, as it accounts for distribution heterogeneity, while hard parameter sharing still performs better than STL.
> >       5. Even when comparing Ablation 2 with Ablation 3—which uses a less effective imputation method—the latter still achieves better predictive performance. This further underscores the advantages of our Step 2 framework over traditional MTL approaches.
> >
> >       Overall, the ablation experiments demonstrate that when both distribution and posterior heterogeneity are present, both steps of our proposed framework are crucial.

---

> > > ### Author Response · Authors · 2024-11-16
> > > **Rebuttal 3 by Authors**
> > >
> > > 4. We have added t-SNE visualization results for the ADNI real data application to better illustrate our method (see **Figure 7, Page 10**). Figure 7 presents the t-SNE visualization of the latent representations obtained from a single training session, where our proposed MTL-HMB method effectively captures both shared and task-specific representations. Notably, the task-specific latent representations of the two tasks display significant differences in their distributions.
> > >
> > > [1] Ye Tian, Yuqi Gu, and Yang Feng. Learning from similar linear representations: Adaptivity, mini-
> > > maxity, and robustness. arXiv preprint arXiv:2303.17765, 2023.
> > >
> > > [2] Doudou Zhou, Tianxi Cai, and Junwei Lu. Multi-source learning via completion of block-wise
> > > overlapping noisy matrices. Journal of Machine Learning Research, 24(221):1–43, 2023.
> > >
> > > [3] Yiming Li, Xuehan Yang, Ying Wei, and Molei Liu. Adaptive and efficient learning with blockwise
> > > missing and semi-supervised data. arXiv preprint arXiv:2405.18722, 2024b.
> > >
> > > [4] Noah Cohen Kalafut, Xiang Huang, and Daifeng Wang. Joint variational autoencoders for multi-
> > > modal imputation and embedding. Nature Machine Intelligence, 5(6):631–642, 2023.
> > >
> > > [5] Jose Bernal, Kaisar Kushibar, Daniel S Asfaw, Sergi Valverde, Arnau Oliver, Robert Mart´ı, and
> > > Xavier Llad´o. Deep convolutional neural networks for brain image analysis on magnetic reso-
> > > nance imaging: a review. Artificial intelligence in medicine, 95:64–81, 2019.
> > >
> > > [6] Fei Xue, Rong Ma, and Hongzhe Li. Statistical inference for high-dimensional linear regression
> > > with blockwise missing data. arXiv preprint arXiv:2106.03344, 2021.

---

### Meta-Review · Area_Chair_Kh1J · 2024-12-16

**Metareview:**

This paper proposes a two-step learning strategy for multi-task learning to address various forms of heterogeneity. It imputes the missing blocks via shared representations that are extracted from homogeneous source across tasks, and disentangles the mappings between input features and responses into a shared component and a task-specific component respectively. Experimental results demonstrate the effectiveness of proposed method.


After discussion, three out of four reviewers are still negative about this manuscript, and concern its evaluation means and experiments. The data generating mechanism is too simple, only a linear model. It remains uncertain whether employing the encoder-decoder framework is more effective than the simple linear imputation used in existing work. Moreover, the role of shared feature extraction encoder and task-specific feature extraction encoder is not clearly presented. So, more efforts could be needed.

**Additional Comments On Reviewer Discussion:**

Three out of four reviewers are still negative about this manuscript, and concern its evaluation means and experiments. The data generating mechanism is too simple, only a linear model. It remains uncertain whether employing the encoder-decoder framework is more effective than the simple linear imputation used in existing work. Moreover, the role of shared feature extraction encoder and task-specific feature extraction encoder is not clearly presented.

---

### Decision · Program_Chairs · 2025-01-22

Reject